

# Evaluation of ARM Tethered Balloon System instrumentation for supercooled liquid water and distributed temperature sensing in mixed-phase Arctic clouds

Darielle Dexheimer[1], Martin Airey[2], Erika Roesler[1], Casey Longbottom[1], Keri Nicoll[2,5], Stefan Kneifel[3], Fan Mei[4], R. Giles Harrison[2], Graeme Marlton[2], Paul D. Williams[2]

[1]Sandia National Laboratories, Albuquerque, New Mexico, USA
[2]University of Reading, Dept. of Meteorology, Reading, UK
[3]University of Cologne, Institute for Geophysics and Meteorology, Cologne, Germany
[4]Pacific Northwest National Laboratory, Richland, Washington, USA
[5]University of Bath, Dept. of Electronic and Electrical Engineering, Bath, UK

*Correspondence to*: Darielle Dexheimer (ddexhei@sandia.gov)

**Abstract.** A tethered balloon system (TBS) has been developed and is being operated by Sandia National Laboratories (SNL) on behalf of the U.S. Department of Energy's (DOE) Atmospheric Radiation Measurement (ARM) User Facility in order to collect in situ atmospheric measurements within mixed-phase Arctic clouds. Periodic tethered balloon flights have been conducted since 2015 within restricted airspace at ARM's Advanced Mobile Facility 3 (AMF3) in Oliktok Point, Alaska, as part of the AALCO (Aerial Assessment of Liquid in Clouds at Oliktok), ERASMUS (Evaluation of Routine Atmospheric Sounding Measurements using Unmanned Systems*)*, and POPEYE (Profiling at Oliktok Point to Enhance YOPP Experiments) field campaigns. The tethered balloon system uses helium-filled 34 m$^3$ helikites and 79 and 104 m$^3$ aerostats to suspend instrumentation that is used to measure aerosol particle size distributions, temperature, horizontal wind, pressure, relative humidity, turbulence, and cloud particle properties and to calibrate ground-based remote sensing instruments.

Supercooled liquid water content (SLWC) sondes using the vibrating wire principle, developed by Anasphere Inc., were operated at Oliktok Point at multiple altitudes on the TBS within mixed-phase clouds for over 200 hours  Sonde-collected SLWC data were compared with liquid water content derived from a microwave radiometer, Ka-band ARM Zenith radar, and ceilometer at the AMF3, as well as liquid water content derived from AMF3 radiosonde flights. The in situ data collected by the Anasphere sensors were also compared with data collected simultaneously by an alternative SLWC sensor developed at the University of Reading, UK; both vibrating wire instruments were typically observed to shed their ice quickly upon exiting the cloud or reaching maximum ice loading. Tethered balloon fiber optic distributed temperature sensing measurements were also compared with AMF3 radiosonde temperature measurements. Combined, the results indicate that TBS distributed temperature sensing and supercooled liquid water measurements are in reasonably good agreement with remote-sensing and radiosonde-based measurements of both properties. From these measurements and sensor evaluations, tethered balloon flights are shown to offer an effective method of collecting data to inform and constrain numerical models, calibrate and validate remote sensing instruments, and characterize the flight environment of unmanned aircraft, circumventing the difficulties of in-cloud unmanned aircraft flights such as limited flight time and in-flight icing.



## 1 Introduction

Understanding microphysical properties of persistent Arctic mixed-phase stratiform clouds is a critical factor in accurately representing the radiative energy balance in climate models (e.g., Morrison et al., 2012; Jouan et al. 2012; Shupe et al., 2013). In particular, supercooled liquid water content (SLWC) within these clouds has great significance in determining the radiation

balance between the surface and clouds (e.g., Shupe and Intrieri, 2004), as well as presenting a potential in-flight icing hazard to aircraft (e.g., Fernandez-Gonzalez et al., 2014). Supercooled liquid water measurements within clouds have been collected using manned aircraft (e.g., Gultepe and Isaac, 1996), but typically not in the Arctic, where operational concerns and the frequent occurrence of these clouds within 2 km of the surface present additional challenges. Surface-based microwave radiometers are widely used to monitor the temporal evolution of liquid water path, i.e. the vertically integrated amount of

liquid water, inside these mixed-phase clouds (e.g., Cremwell et al., 2009). Liquid water has no absorption line in the microwave spectrum, however, so these instruments cannot directly provide information on the distribution of the SLWC vertically inside the cloud, which is key for radiation and ice microphysics. Development of a tethered balloon system was supported by the U.S. Department of Energy's (DOE) Atmospheric Radiation Measurement (ARM) program in order to collect semi-regular, in situ measurements within Arctic clouds while avoiding the expense and potential risk of manned aircraft

flights.

Tethered balloon systems have been used to collect tropospheric atmospheric measurements for over 40 years, including profiles of biogenic compounds, chemical species, turbulence, radiation, and meteorological parameters (e.g., Morris et al., 1975; Owens et al., 1982; Greenberg et al., 1998; Knapp et al., 1998; Egerer et al., 2019). Morris et al. (1975) developed a portable tethered balloon system that essentially behaved as a tethered radiosonde that was able to be operated by one person

at altitudes up to 750 m above ground level (AGL) in wind speeds as high as 10 m/s. Owens et al. (1987) advanced the capabilities of tethered balloon systems by creating a system capable of lifting 2.75 kg to 800 m AGL that was used to collect meteorological data and ozone concentrations. Greenberg et al. (1998) further promoted tethered balloon system development by deploying sampling packages used to measured biological volatile organic compounds in the mixed layer in a series of deployments conducted over 11 years. Knapp et al. (1998) combined tethered balloon system and kite measurements to study

the anticorrelation between ozone and water vapor mixing ratios. Most recently, in 2019 Egerer et al. (2019) operated sets of instruments to measure turbulent, energy, and radiative fluxes to altitudes of 1.5 km AGL within Arctic clouds.

Use of tethered balloon systems can be limited by the very meteorological conditions that would be desirable to operate during however, including elevated wind speeds and wind speed and directional shear, and convective updrafts and downdrafts. Additionally, outside of restricted airspaces tethered balloon systems are often unable to receive aviation authority approvals

in the U.S. to operate near or within clouds, to altitudes higher than 1 km above ground level, or in reduced surface visibilities. The work discussed herein pertains to the new capability of using tethered balloon systems within restricted airspace for persistent flight inside Arctic mixed phase clouds, with supercooled liquid water sondes and distributed temperature sensing optical fiber systems.



Vibrating wire-based devices for measuring supercooled liquid water on radiosondes have been in development since the 1980s (e.g., Hill and Woffinden, 1980; Hill, 1994). In the past decade vibrating wire-based supercooled liquid water content radiosonde flights have been conducted concurrently with a collocated microwave radiometer, ceilometer, and Ka-band radar to validate the sonde-measured vertical profile of supercooled liquid water (e.g., Serke et al., 2014; King, 2016). Advancing

this approach, supercooled liquid water content sondes from two manufacturers were operated on the ARM TBS at multiple altitudes within Arctic clouds simultaneously for over 200 hours, in order to collect comparatively higher spatially and temporally-resolved data than were available from radiosonde balloon flights. Supercooled liquid water measurements from collocated sondes from one manufacturer, that were operated simultaneously on the TBS, were used to estimate the measurement uncertainty. Liquid water path from the zenith-pointing microwave radiometer at Oliktok Point was adiabatically

distributed through a single cloud layer using the ceilometer-determined cloud base and Ka-band radar-determined cloud top altitudes for inter-comparison.

The microwave radiometer does not discriminate between liquid and supercooled liquid, and is insensitive to ice and snow at frequencies lower than 90 GHz, so a high-resolution temperature profile is desirable when conducting comparisons of SLWC sonde and radiometer measurements within Arctic clouds composed of cloud water in both conditions; where water that

continues to exist in a liquid state at temperatures below 0 °C is considered supercooled. In addition to radiosonde-based measurements of temperature from each SLWC sonde, near-continuous measurements of temperature were collected using a fiber optic distributed temperature sensing (DTS) system. DTS provided vertical profiles of temperature every 0.25 m between the surface and the balloon every 30 to 60 s. Distributed temperature sensing has been shown to be an effective method of collecting atmospheric temperature measurements (e.g., Keller et al., 2011, Thomas et al., 2012, de Jong et al., 2015), but has

been limited in the duration, altitude, and ambient conditions of measurement. The present paper discusses tethered balloon-based distributed temperature sensing measurements and their comparison with concurrent AMF3 radiosonde temperature measurements under cloudy and clear conditions to altitudes over 1 km above the surface. An overview of the tethered balloon system is provided, followed by descriptions of the SLWC sondes and DTS system, subsequent flight results from Oliktok Point, and a discussion of future operational plans.

**2 Tethered Balloon System (TBS)**

**2.1 TBS components**

The TBS may be driven mechanically by a 2 horsepower DC motor and reversible variable speed controller or smaller electrical winches depending on the mission and balloon in use. The most commonly used winch deploys over 2 km of Plasma® 12 strand synthetic rope, which has a minimum breaking strength of 2,494 kg (Cortland Company, 2019). Allsops 34 m³ helikites

(Fig. 1) are a balloon/kite hybrid that use lighter-than-air principles to obtain initial lift, and then a kite to achieve stability and dynamic lift. Helikites are typically used for flights with desired altitudes to 700 m Above Ground Level (AGL), a maximum payload of less than 10 kg, and in surface wind speeds less than 11 m/s. SkyDoc™ and Drone Aviation Corp 79 – 104 m³ aerostats use a skirt to maintain orientation and stability in flight.



Aerostats are generally used when the desired maximum flight altitude is higher than 600m, the payload is 10 – 25 kg, and in surface wind speeds less than 8 m/s (see Dexheimer (2018) for a full description of the TBS).

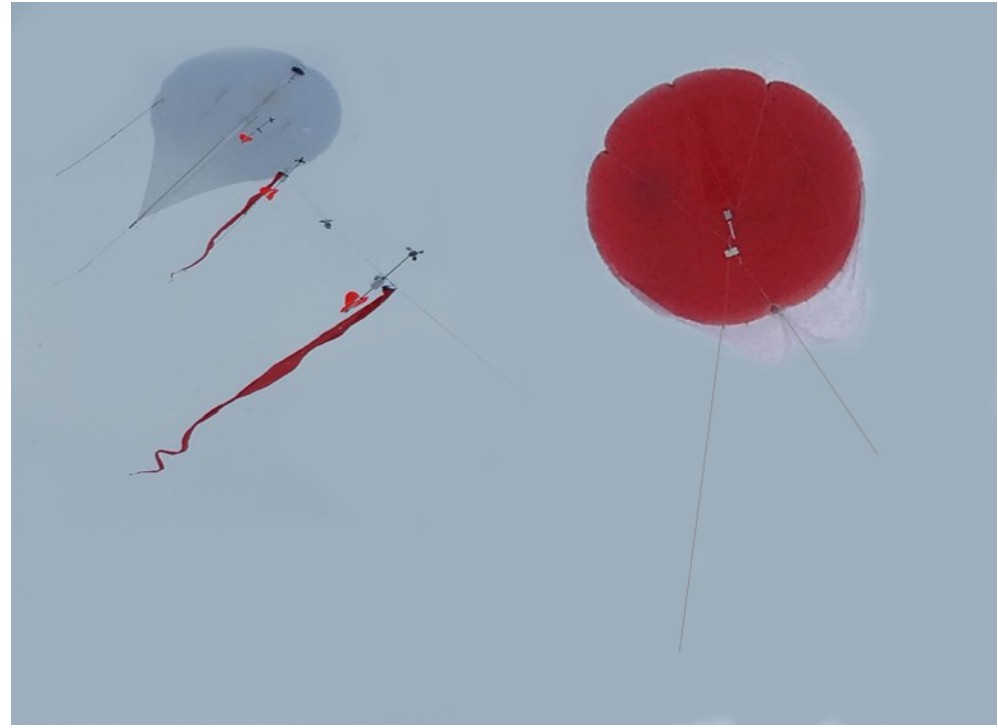

**Figure 1.** 34 m$^3$ helikite in flight with three tethersondes (left) and 79 m$^3$ aerostat in flight with radar calibration sphere (right).

5  **2.1.2 TBS operations**

The TBS was operated with multiple instrument payloads. This paper focuses on TBS flights using SLWC sondes and DTS at the AMF3 at Oliktok Point, Alaska, for almost 337 hours from October 2015 – September 2018. Flights occurred during daylight to altitudes of 1.45 km AGL and with durations from one to nine hours in various atmospheric conditions including clear sky, broken to overcast clouds, rain, sleet, snow, and temperatures from -20 °C to 25 °C.

10  **Table 1.** Overview of TBS flights analyzed within this study including date, duration, sensor payload, and campaign.

| Dates | TBS Flight Hours | Relevant Sensors | Campaign |
|---|---|---|---|
| October 22-28, 2015 | 33.5 | SLWC sondes | ERASMUS |
| April 3-20, 2016 | 9.3 | SLWC sondes | AALCO, ERASMUS |
| May 13-16, 2016 | 14.8 | SLWC sondes, Sensornet Oryx DTS | AALCO, ERASMUS |





| June 5-11, 2016 | 24.0 | SLWC sondes, Sensornet Oryx DTS | AALCO, ERASMUS |
|---|---|---|---|
| July 24-27, 2016 | 7.4 | Sensornet Oryx DTS | AALCO, ERASMUS |
| October 10-20, 2016 | 33.0 | SLWC sondes, Sensornet Oryx DTS | AALCO, ERASMUS |
| November 14-17, 2016 | 10.5 | SLWC sondes | AALCO |
| April 2-10, 2017 | 8.5 | SLWC sondes | AALCO, ERASMUS |
| May 15 – 24, 2017 | 30.8 | SLWC sondes, Sensornet Oryx DTS with Fiber Optic Rotary Joint (FORJ) | AALCO, ERASMUS |
| August 4 – 9, 2017 | 17.0 | SLWC sondes, Sensornet Oryx DTS with Fiber Optic Rotary Joint (FORJ) | AALCO, ERASMUS |
| October 13 – 22, 2017 | 9.7 | SLWC sondes, Silixa XT DTS with Fiber Optic Rotary Joint (FORJ) | AALCO, ERASMUS |
| July 1 – 11, 2018 | 41.8 | SLWC sondes, Silixa XT DTS with Fiber Optic Rotary Joint (FORJ) | POPEYE |
| July 24 – August 3, 2018 | 43.5 | SLWC sondes, Silixa XT DTS with Fiber Optic Rotary Joint (FORJ) | POPEYE |
| August 17 – 26, 2018 | 22.9 | SLWC sondes, Silixa XT DTS with Fiber Optic Rotary Joint (FORJ) | POPEYE |
| September 21 - 28, 2018 | 29.5 | SLWC sondes, Silixa XT DTS with Fiber Optic Rotary Joint (FORJ) | POPEYE |
| **TOTAL** | **336.2** | | |

**2.2 TBS Anasphere SLWC sondes**

SLWC sondes developed by Anasphere Inc. were operated on the TBS with both InterMet (iMet) radiosondes and Anasphere
tethersondes (Fig. 2). The vibrating wires on the SLWC sondes were oriented orthogonal to the freestream direction, meaning
they were oriented perpendicularly to the surface on the TBS. The rate of change of the frequency of the 0.61 mm diameter
steel vibrating wire on the SLWC sonde and other atmospheric parameters were used to calculate supercooled liquid water
based on Equation 1, where $b_0$ is the vibrating wire mass per unit length of 2.24 g m$^{-1}$, $f_0$ is the un-iced wire frequency in Hz,
$f$ is the wire frequency in Hz at time $t$, $\epsilon$ is the droplet collection efficiency between 0 and 1 found using the method described
in Lozowski et al. (1983), D is the wire diameter in m, and $\omega$ is the velocity of the air relative to the wire in m s$^{-1}$.

$$SLWC = -\frac{2b_0{f_0}^2}{\epsilon D \omega f^3}\frac{df}{dt} \qquad (1)$$





The raw wire frequencies had outliers removed if the frequency deviated over 0.1 Hz from a 30s moving average of the frequency, and the remaining frequencies were then smoothed using the robust LOESS (locally estimated scatterplot smoothing) model. Wind speeds from the Doppler lidar at the AMF3 or tether-based anemometers were used in the calculation. Pressure, temperature, and relative humidity values from iMet radiosondes were typically used in the collection efficiency

5      calculation, if radiosonde measurements were unavailable tethersonde-measured values of these parameters were used. An estimate of median droplet diameter, $d_0$, was required for the collection efficiency calculation. SLWC was calculated using median droplet diameters of 11, 16, and 20 μm based on Lozowski et al. (1983) and Bain and Gayet (1981), with results for a median droplet diameter of 16 μm being presented here. At wind speeds ≥ 5 m/s, which were typical during TBS flights, a median droplet diameter of 16 μm results in a collection efficiency greater than ~0.9. Therefore, we use this diameter to get

10     the lower estimate of SLWC in all deployments so as provide the most conservative estimates given our lack of particle size knowledge.    The three median droplet diameters had limited impact on the resulting calculated SLWC, with mean SLWC values for each TBS flight being within +/- 0.01 g/m$^3$ when all other values were kept constant and the median droplet diameter was varied. A full discussion of the Anasphere SLWC sonde measurement theory and design is available in Serke et al., 2014 and King, 2016.

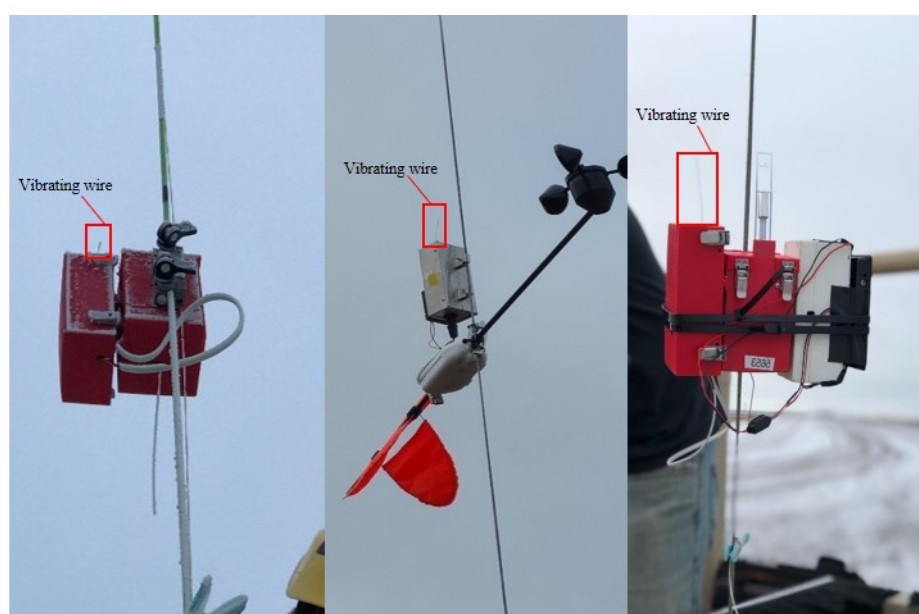

**Figure 2**. Anasphere SLWC sonde left of InterMet radiosonde on TBS tether (left). Anasphere SLWC sonde above Anasphere tethersonde (center). From left-to-right Anasphere SLWC sonde, InterMet radiosonde, Reading SLWC sonde on TBS tether (right).

Another SLWC detector, developed at the University of Reading, UK (Airey et al. 2017), was operated alongside the Anasphere sensor on some of the deployments to provide independent comparison and validation. This sensor was designed with programmable versatility in mind. It was also designed with disposability for routine radiosonde use, by implementing





complex on-board processing on relatively cheap hardware. The device operates on the same principle as the Anasphere sensor, that is, a vibrating wire that determines mass accretion (ice) from a reduction in natural oscillation frequency, however it is highly versatile, with programmable on-board processing that measures the frequency in three different ways. Implemented methods to determine the resonant frequency include a Fast Hartley Transform (FHT), a frequency sweep, and a Phase-Locked

Loop (see Airey et al. (2017) for a full description of these methods). This experiment combined the FHT and frequency sweep modes, the former providing fast identification of the broad region of frequency ($\pm$0.2 Hz), the latter using this to focus the sweep region for more rapid resonance detection, which also provides a much higher precision ($\pm$0.005 Hz). Outlier removal and data smoothing is also required for this sensor, in this case fitting method, that uses a first order polynomial, achieves a better fit to the data given the longer update time when compared with the Anasphere sensor, which uses the second order

polynomial fit; both implementations use the fitting models defined in Mathworks' MATLAB version 2018b.  The SLWC is calculated using the same method as that for the Anasphere device. In contrast to the Anasphere design, which mechanically actuates the wire, the Reading design uses a piezoelectric device to both drive the wire and measure the frequency after the drive ceases, thereby eliminating all moving parts. The sensors were flown with the collecting wire horizontal, to better sense the lateral airflow. This sensor is designed to relay the data through a radiosonde via the PANDORA interface, also developed

at Reading (Harrison et al. 2012). In normal operation the data are transmitted via the standard radiosonde telemetry; for this study, it was adapted to function as a standalone unit, self-logging to SD card.

### 2.2.2 Anasphere SLWC sonde uncertainty

Throughout the ICARUS and AALCO campaigns pairs of eight different Anasphere SLWC sondes were operated side-by-

side, in the presence of SLW clouds, for over four hours.  Three such comparison flights were conducted with SLWC sondes on the TBS, while one flight was conducted using a free-flight meteorological balloon.  The SLWC values calculated at simultaneous times for each SLWC sonde pair are shown in Figure 3. The mean differences between simultaneous non-zero SLWC values calculated by all sonde pairs operated on the TBS were 0.01 to 0.02 g/m$^3$, and larger for the free-flight balloon pair at 0.06 g/m$^3$.





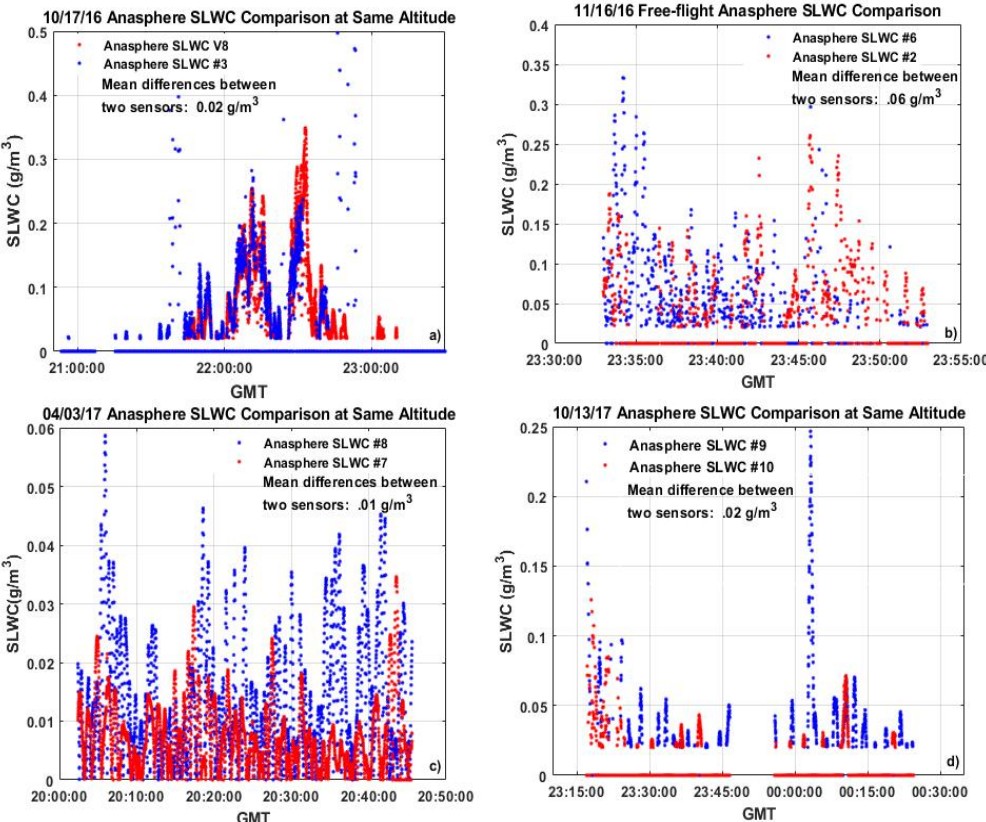

**Figure 3.** Results of four side-by-side comparison flights of Anasphere SLWC sondes.

## 2.3 Distributed Temperature Sensing system components

Two DTS systems were used on the TBS over the three field campaigns studied. The Sensornet Oryx DTS fires a center

5   wavelength 971 nm laser pulse lasting less than 10 ns through attached 50 µm multimode optical fibers. Up to four fibers may

be deployed from each DTS system simultaneously. Some of the laser light is Rayleigh scattered as it collides with the

structure of the fiber and returns down the fiber at the incident wavelength. The portions of the backscattered signal that are

shifted towards lower and higher frequencies are considered Stokes and anti-Stokes scattering, respectively. The ratio of Anti-

Stokes to Stokes photons produced increases with temperature, and their different respective attenuations are combined in

10   order to represent the proportional total return intensity. The velocity of light in the fiber is constant, so the number of ns

between the laser pulse firing and the detection of the returned light can be used to determine the scattering site, and thus the

calculated temperature. When DTS is operated on the TBS the scattering site represents an altitude. Under Equation 2 below,

the intensity of the backscattered light (I) is proportional to the difference in the molecular energy state of the photons before

and after scattering ($\Delta E$) divided by the Boltzmann constant (k) and the temperature of the scattering site.





$$\frac{I_{Stokes}(z)}{I_{antiStokes}(z)} \alpha \exp(\Delta E / kT(z))$$

(2)

When the balloon is stationary DTS data may be collected by directly connecting an optical fiber to the DTS system. However, there are some disadvantages of this configuration for the TBS: at least twenty minutes are required to install and remove coils of fiber in calibration baths, there is a potential risk of damage to the fiber whenever it is coiled or uncoiled, and the TBS is

required to float at a fixed altitude when vertical profiling may be a more desirable method of operation. To overcome these constraints DTS data may be collected when the balloon is in motion by using a fiber optic rotary joint (FORJ) between the optical fiber and DTS. However, the low loss (< 0.5 dB) required for DTS measurements approaches the limits of most currently-available commercial FORJs. Multiple FORJs were tested before successfully collecting accurate DTS measurements through an FORJ by using a spool of fiber deployed with a variable-speed electric motor. The fiber was spooled

and unspooled using foot pedals to match the rate of the TBS winch during ascent and descent (Fig. 7). If a significant temperature differential does not exist between the surface and lowest few meters of the atmosphere, a method of demarcating the surface is helpful in determining the starting location of the suspended portion of fiber with respect to the portion of fiber remaining on the spool. Various methods of surface demarcation were tested and a saltwater bath proved to be the most ideal solution.

A Sensornet Oryx DTS system was used prior to October 2017, with a 30 s measurement interval and 1 m sampling resolution. Single-ended DTS measurements were collected after initially collecting double-ended measurements, due to the reduction in datafile size and processing effort related to correlating the deployed fiber length with the balloon altitude, as it is affected by horizontal drag. Approximately 15 m of fiber were coiled into ice water and hot water calibration baths, with 15 m of fiber between each bath. A PT100 temperature sensor was placed in each bath and logged by the DTS. An iMet-1-RSB radiosonde

measuring temperature every 1 s was placed at the balloon-end of the fiber to serve as an independent temperature measurement aloft for calibration. In September 2017 a second DTS system, a Silixa XT, was procured. The Silixa XT has a center wavelength of 1064 nm and is capable of 25 cm spatial resolution, largely due to a reduction in pulse duration to 2.5 ns, which allowed a smaller section of fiber to be used in the surface calibration baths and demarcation portion.





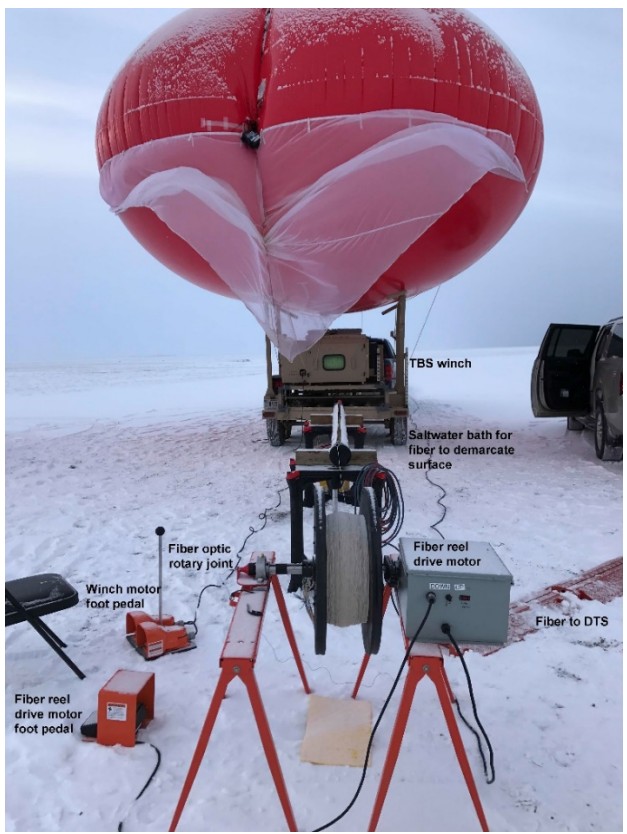

**Figure 4.** TBS with optical fiber operating through rotary joint and saltwater bath

## 3 Results

### 3.1 SLWC Results

#### 3.1.1 10/13/17 SLWC from TBS flight with concurrent SLWC from free radiosonde launch

Between 10/13/17 22:20 UTC and 10/14/17 00:20 UTC two Anasphere SLWC sondes were operated in the presence of two stratocumulus cloud layers, the lowest with a base at 0.45 - 0.55 km and a second with a base at approximately 0.75 km and a top near 1 km. These cloud layers were representative of the persistent, low-level stratocumulus clouds which commonly occur in the Arctic. A stratocumulus cloud base between 100 m and 1.2 km persisted at Oliktok Point for 96 hours between 10/12/17 and 10/16/17. Temperature decreased during the TBS flight from -2 °C at the surface to -5.5 °C near 600 m.

The ARM AMF3 23:27 UTC sounding, that occurred during the TBS flight, was analyzed using the commercial software RAOB (Figure 5). LWC was calculated from the sounding using the enthalpy equation for cloud water (LWC) in RAOB. This equation uses the adiabatic Enthalpy (gamma) lapse-rate equation, where LWC is a function of air density $\rho(z)$, specific heat at constant pressure ($Cp$), latent heat of vaporization (L), dry adiabatic lapse rate ($\Gamma d$), and the moist adiabatic lapse rate ($\Gamma s$).





$$LWCad(z) = \int \rho(z)\frac{Cp}{L}(\Gamma - \Gamma s)dz \tag{3}$$

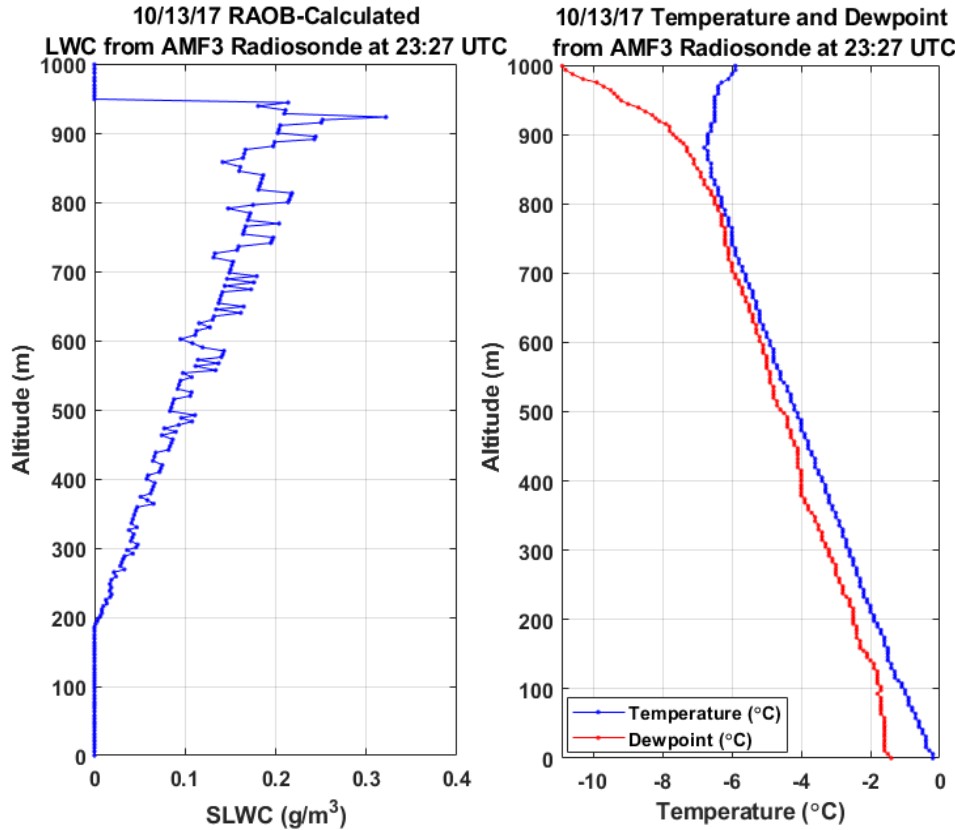

**Figure 5:** LWC calculated in RAOB software using Equation 3 for 10/13/17 23:30 UTC AMF3 sounding (left) and temperature and dewpoint from sounding (right).

The LWC calculated by RAOB, which in this case was considered SLWC since the entirety of the cloud was below 0 °C, increased adiabatically through the cloud reaching a maximum of 0.32 g/m³ just below cloud top at 0.95 km. Supercooled liquid water content was also calculated from the two SLWC sondes operating on the TBS. The LWC values calculated by RAOB from the free radiosonde flight at the same altitudes as both tethered balloon vibrating-wire SLWC sondes were both 0.14 g/m³. The lowest cloud base reported by the AMF3 ceilometer between 23:26 and 23:32 UTC had a standard deviation of 120 m and varied widely from a minimum of 210 m to a maximum of 740 m. This variation in the cloud base would be expected to cause significant variation in whether or not SLWC was measured by the TBS SLWC sondes. The maximum SLWC observed by the highest altitude TBS SLWC sonde between 23:26 UTC and 23:32 UTC was 0.14 g/m³, while the maximum SLWC observed by the lower altitude sonde was 0.05 g/m³. Given the variation in the cloud base during the flight and the spatial variation between the TBS and AMF3 radiosonde measurements, the TBS SLWC sondes and RAOB LWC calculation showed reasonable agreement.



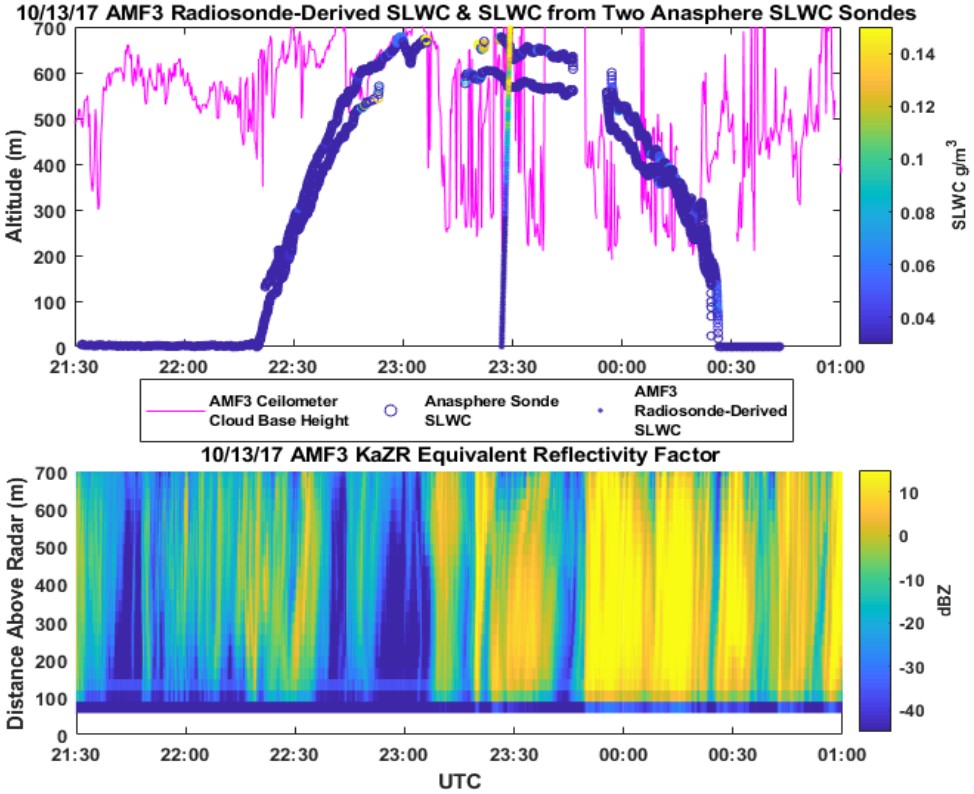

**Figure 6:** TBS Flight of two SLWC sondes with concurrent free balloon radiosonde launch at 10/13/17 23:27 UTC. The RAOB LWC values calculated from the 23:27 UTC AMF3 free radiosonde launch are plotted (dots), as well as SLWC measured by two TBS SLWC sondes (circles), the lowest cloud base height reported by the AMF3 ceilometer (magenta), and the reflectivity from the AMF3 KaZR (bottom plot).

### 3.1.2 10/15/16 and 10/20/16 TBS Anasphere SLWC sondes and SLWC from MWR

For two TBS flights that did not occur during one of the twice daily AMF3 radiosonde launches, SLWC values measured by TBS Anasphere SLWC sondes were compared with SLWC derived from the surface-based AMF3 MWR. These TBS flights occurred in single-layer, subfreezing stratocumulus clouds on 10/15/16 and 10/20/16. SLWC was derived by distributing MWR Liquid Water Path values adiabatically through the single cloud layer. The cloud layer thickness was defined using the lowest cloud base from the ARM AMF3 ceilometer and cloud top from the ARM ARSC (Active Remote Sensing of Clouds) Value-Added Product (VAP). The ARSC VAP combines ceilometer data and the deviation of the KAZR reflectivity from received sky noise to assign bases and tops to up to 10 cloud layers. Non-zero SLWC values $\leq 0.02$ g/m$^3$ were considered to be below the noise threshold of the Anasphere SLWC sonde and removed, then all remaining SLWC values were smoothed with a moving average.

On 10/15/16 the cloud base altitude deviated significantly with time, resulting in the Anasphere SLWC sondes flying above and below the ceilometer-defined cloud base (Figure 7). The SLWC values from both sensors were non-zero when in-cloud





as expected although the magnitude differed, resulting in an $R^2$ value of 0.38. Given the ceilometer cloud height resolution of +/- 10 m (e.g., Morris, 2016) and TBS iMet radiosonde GPS altitude resolution of +/- 15 m, the agreement of SLWC detection between the two sensors is surprisingly good considering the uncertainty regarding the placement of the SLWC sonde with respect to cloud base.

5   On 10/20/16 the Anasphere SLWC sonde flew for two hours at 150 m above cloud base, descending to 85 m above cloud base with time, due to the accumulation of ice on the balloon, sensors, and tether. The Anasphere SLWC sonde experienced multiple shedding events during the flight, where the maximum ice load on the vibrating wire is reached and subsequently shed, resulting in erroneously low SLWC values. The $R^2$ value for SLWC values from the flight was 0.79, with MWR SLWC values averaging 0.03 g/m³ higher, largely due to offset low sonde SLWC values during shedding events.

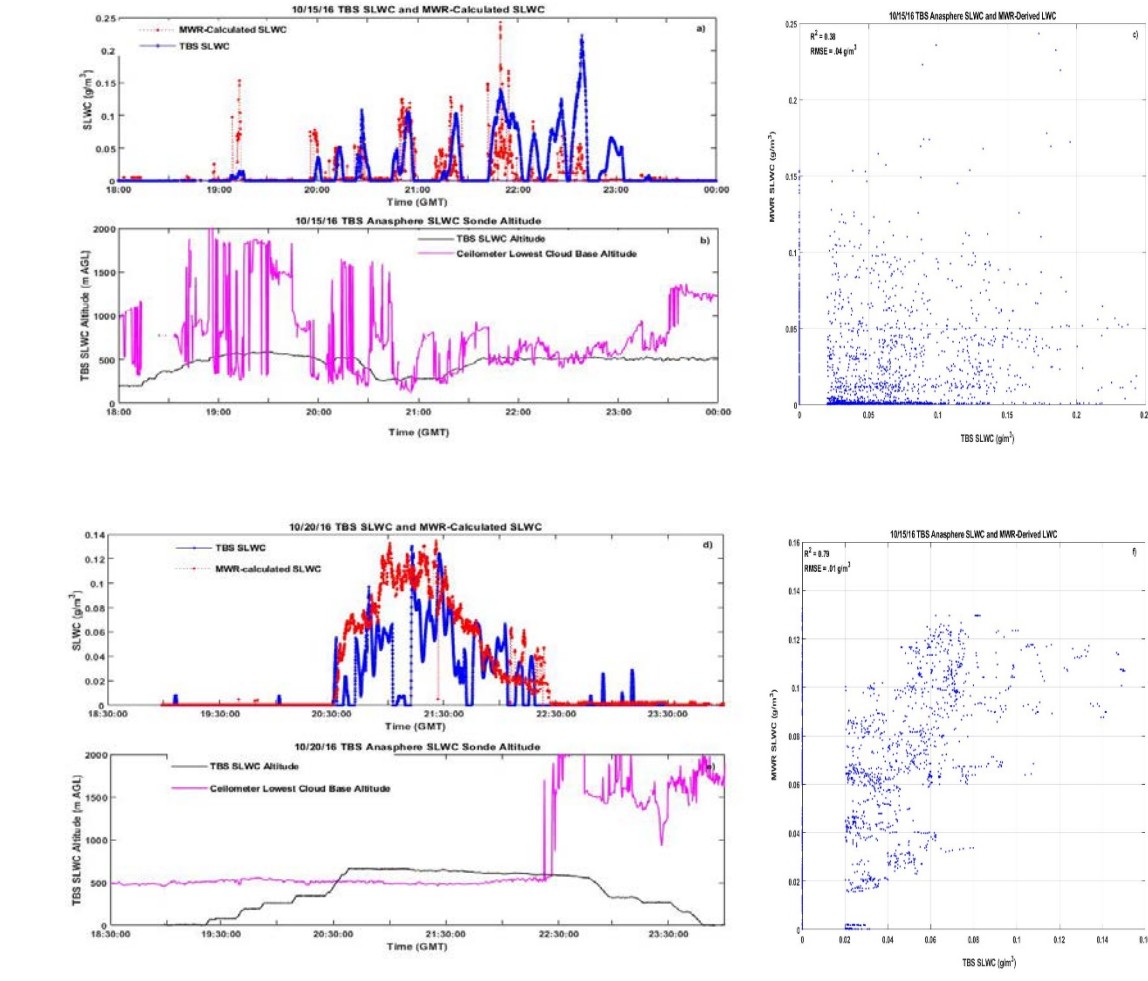

15   **Figure 7:** 10/15/16 SLWC from TBS Anasphere sonde and calculated from MWR (a), 10/15/16 TBS Anasphere sonde and ceilometer cloud base altitude (b), and 10/15/16 SLWC from Anasphere sonde vs. SLWC calculated from MWR (c), 10/20/16 SLWC from TBS Anasphere sonde and calculated from MWR (d), 10/20/16 TBS Anasphere sonde and ceilometer cloud base altitude (e), and 10/20/16 SLWC from Anasphere sonde vs. SLWC calculated from MWR (f).





### 3.1.3 Comparison of simultaneous in situ SLWC measurements from the Anasphere and Reading sensors on the TBS

To independently test the validity of the in situ measurements collected by the Anasphere sondes, some balloon flights were instrumented with an additional Reading SLWC sensor so that simultaneous profiles could be taken and compared between

the two sensing methods and against the MWR data. One such flight was conducted on 08/02/18, where the sensors were deployed on the helikite platform through the cloud base to an altitude of ~400 m and returned to ~150 m below the cloud base over two cycles, each of around 60 minutes duration. Both sensors successfully detected SLW, particularly during the ascent and descent phases at approximately 2100, 2130, 2200, and 2230. During the initial ascent a gradual increase in SLWC is observed between 200 – 400 m from 0 – 0.3 g/m$^3$. At the maximum altitude, SLWC decreased to 0 g/m$^3$ as the sensors emerge

into a region of low relative humidity (~65%) interpreted to be above the cloud top. During the subsequent descent, both sensors once again detected similar values of SLWC, albeit lower, probably due to either the vibrating wires being at maximum ice loading or descent through an anomalously low-SLW region. The MWR LWP detected during this descent (Figure 9) would suggest the former. The second cycle follows the same pattern, with similar SLWC during ascent and above the cloud, although the Reading sensor detected somewhat less SLW during this ascent possibly due to the retention of more ice on the

wire than the Anasphere sensor. The final descent also shows reasonably good agreement, peaking at ~0.3 g/m$^3$ between 300-350 m as observed in the preceding ascents. Following the final descent, both sensors continue to detect SLW while they are held at ~150 m, somewhat higher than the period following the first descent and, in this case, coinciding periodically with the cloud base, which would account for the continued detection. The higher LWP value detected by the Reading sensor during descent 2 may be due to the longer sensing wire (120 mm rather than 90 mm) allowing a greater overall detection possible

after the shorter wire reaches maximum ice loading, which in King 2016 is suggested as 0.75 g/m$^3$ as the response of the wire becomes non-linear in that environment.

The calculated LWP from each sonde is compared with that derived from the MWR in Figure 9 and show both sensors achieving good agreement with the MWR data during the ascents (to within ±0.008 mm for the Reading sensor and ±0.003 mm for the Anasphere sensor); however, this is less good during the descents for the possible reasons discussed above. Figure

8 shows the time series of the flight and Figure 9 compares the two SLWC profiles.




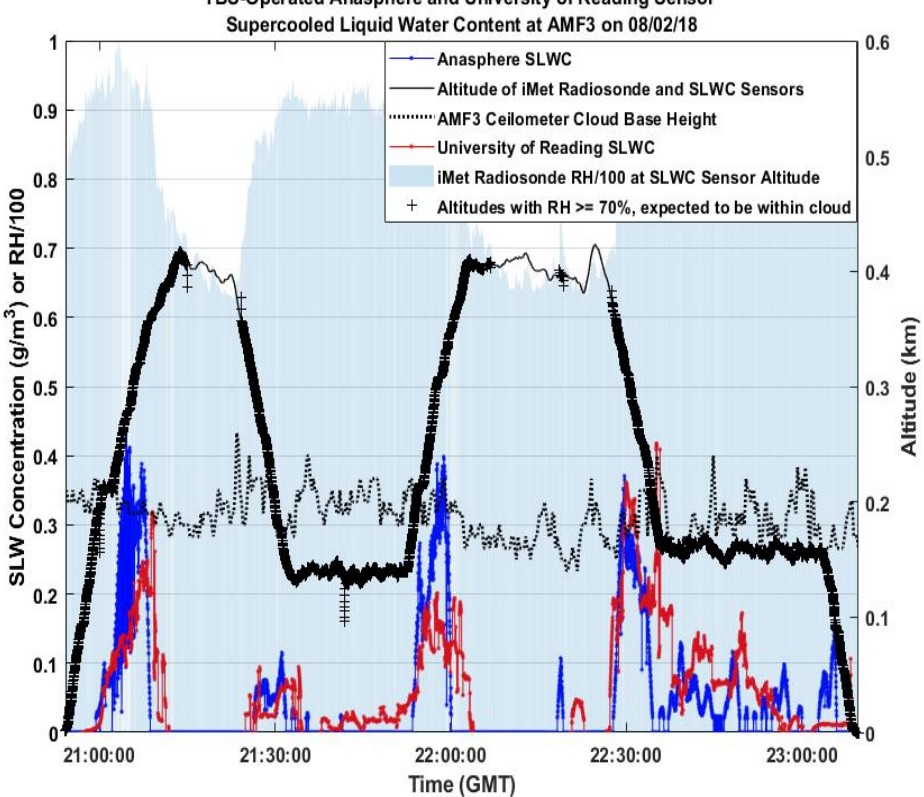

**Figure 8:** Time-series of 08/02/18 TBS flight with Anasphere and Reading SLWC data, sensor altitude and relative humidity, and cloud base height.





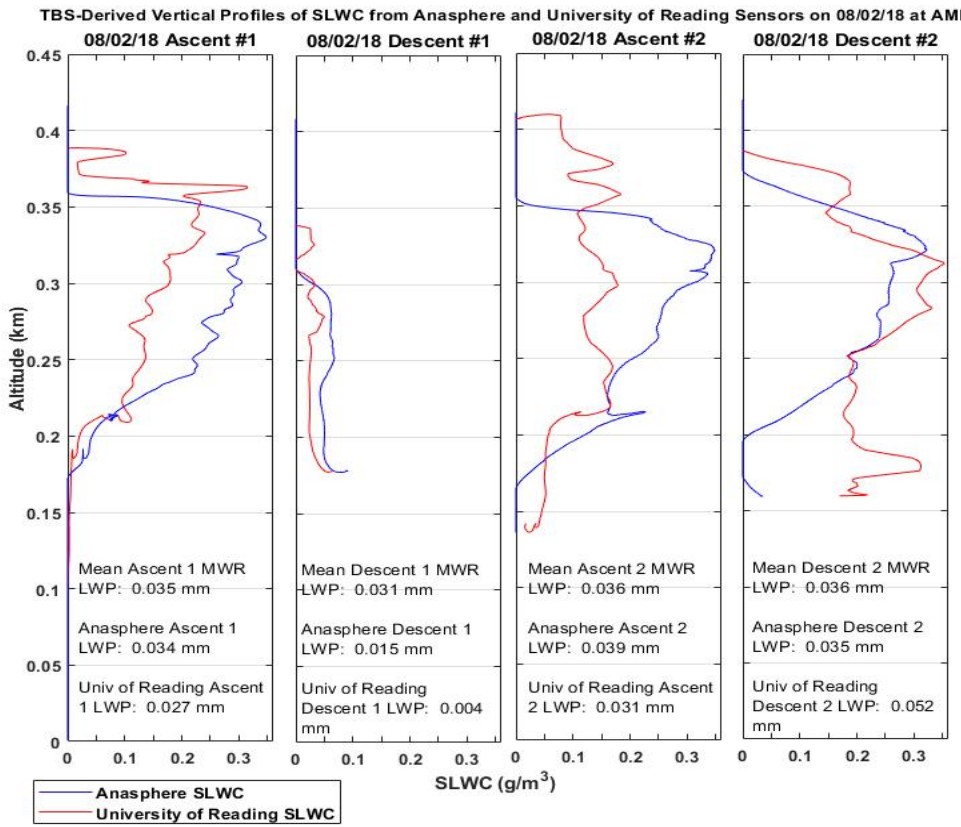

**Figure 9:** Vertical profiles of Anasphere and Reading SLWC from 08/02/18 TBS flight with mean surface-based MWR LWP values.

Overall, the two sensors seem to provide broadly similar SLWC profiles, but not without some discrepancies. This increases confidence in the use of each of them and provides independent verification of the measurements. The variation between them

5   is partly due to both methods being reliant on different sensor data acquisition methods and physical geometry, but also the sampling conditions of the two sensors may differ. The conversion of the frequency of the wire oscillation to SLWC is non-trivial depending on the appropriateness of theoretical assumptions, and small differences in frequency data may result in somewhat larger discrepancies in the derived SLWC. The processing of outliers and smoothing of data to allow df/dt to be obtained is another source of sensor to sensor variation but, given these considerations, the good agreement between them

10  provides some confidence in their measurements. The Reading sensor has higher precision and sampling frequency than the Anasphere sensor, but is more prone to data gaps. Therefore, the result of the processing algorithms applied will result in sensor-specific nuances.

### 3.1.4 Comparison of in situ SLWC Anasphere sonde measurements and SLWC calculated from radiosonde flights




Mean values of SLWC from the in situ vibrating wire sondes deployed on the TBS for 43 flights were compared with mean SLWC values calculated from AMF3 radiosonde launches that occurred during each TBS flight at the altitudes of the TBS SLWC sondes using the enthalpy lapse-rate equation for cloud water (LWC) in RAOB shown in Equation 3.

The mean SLWC values measured by the in situ vibrating wire sondes averaged 0.045 g/m$^3$ higher than the mean SLWC values calculated from the radiosonde flights using Equation 3. Some of this difference could be attributed to temporal and spatial variation between the TBS and radiosonde flights. Despite this however, the mean SLWC values calculated from Equation 3 are quite small, at < 0.05 g/m$^3$ for 91% of the dataset. Previous aircraft measurements in supercooled stratiform clouds measured SLWC values < 0.05 g/m$^3$ for 36% of the samples (e.g., Sand et al., 1984), which is consistent with the results from the TBS Anasphere sonde measurements at 38%.

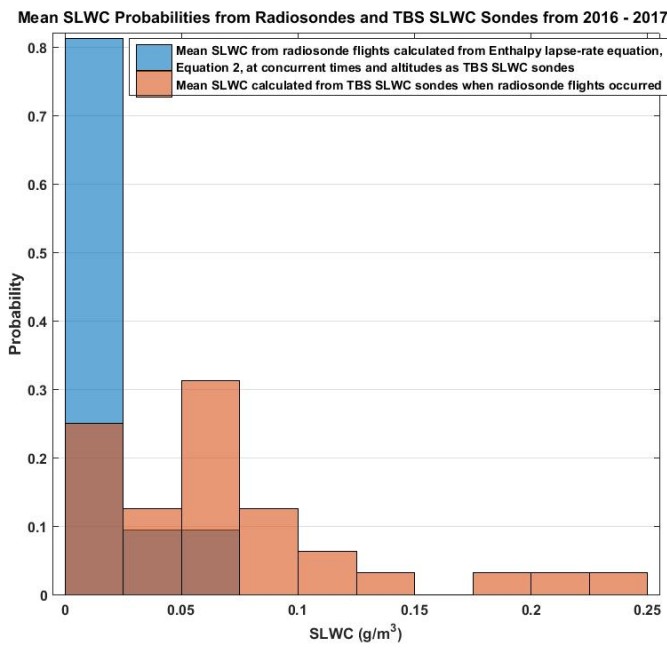

**Figure 10:** 2016– 2017 Mean SLWC from Anasphere sondes (orange) and mean SLWC calculated from AMF3 radiosonde flights (blue) during TBS flights over the TBS flight altitude range.

### 3.1.5 SLWC from TBS Anasphere sondes from 2015 – 2017 by month, altitude, and temperature

Recurring TBS Anasphere SLWC sonde deployments occurred at the ARM AMF3 during fall and spring months between 2015 and 2017. The highest SLWC values were measured in the late spring during May and June, with lower values being measured in fall and early spring. Measured SLWC values increased at flight altitudes between 400 m and 1 km AGL and were lower below 400 m. The highest measured SLWC values occurred at temperatures above -14 °C and below -2 °C. In respect to interannual variability of SLWC, the mean SLWC values in three sequential Octobers were 0.06, 0.10, and 0.14 g/m$^3$; sequential average April values were both 0.05 g/m$^3$, and sequential May means were 0.26 and 0.14 g/m$^3$, respectively.





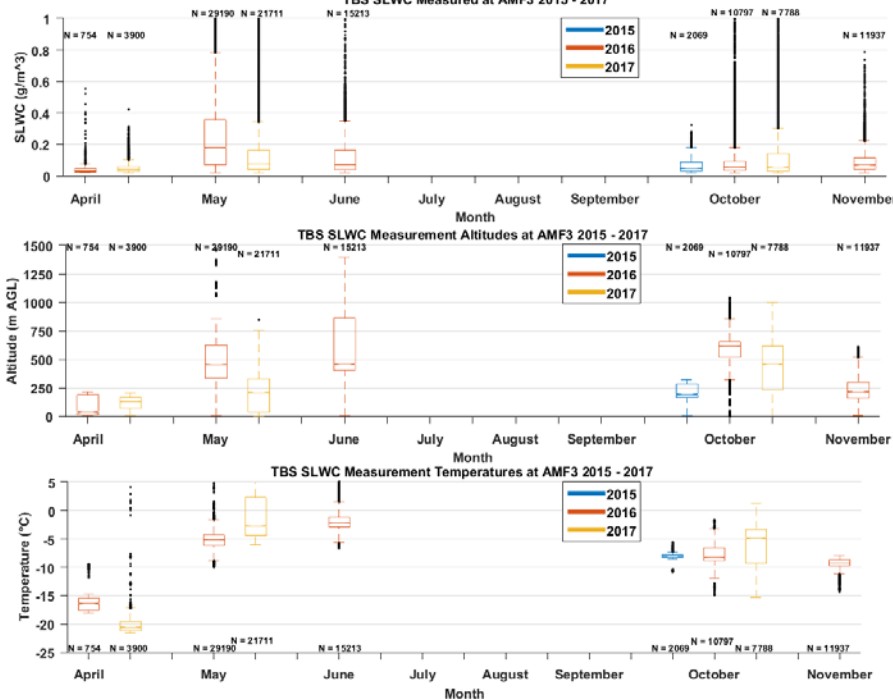

**Figure 11:** SLWC measured by Anasphere sondes flown on the TBS and SLWC measurement altitudes and temperatures by month from 2015 – 2017, where N refers to the number of SLWC measurements and the median is shown by the horizontal line. The 25th and 75th percentiles are shown by the bottom and top edges of the box, the whiskers extend to the data points not considered outliers, and outliers are plotted with '+'.




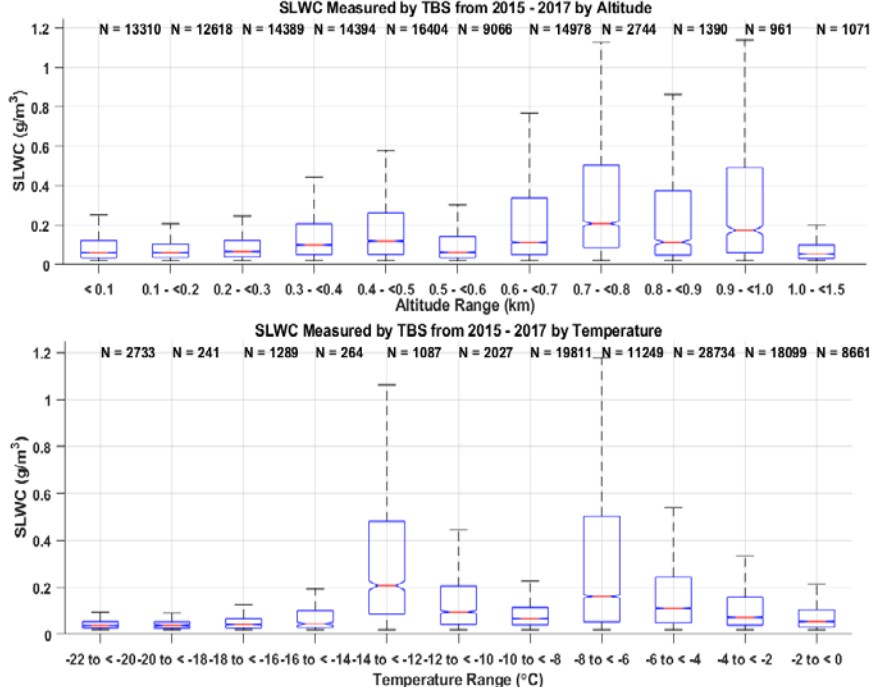

**Figure 12:** SLWC measured by Anasphere sondes flown on the TBS by altitude and temperature from 2015 – 2017, where N refers to the number of SLWC measurements and the median is shown by the horizontal line. The 25th and 75th percentiles are shown by the bottom and top edges of the box, the whiskers extend to the data points not considered outliers, and outliers are plotted with '+'.

## 3.2 DTS Results

### 3.2.1 6/11/16 DTS measurements with concurrent free radiosonde launch

DTS measurements were collected once a minute with a Sensornet Oryx using fiber suspended along the TBS tether from 6/11/16 21:18 – 6/12/16 01:19 UTC. During this time a free flight radiosonde was launched from the AMF3 at 23:30 UTC. The vertical resolution of the radiosonde was approximately 10 m, while DTS measurement vertical resolution was every 1 m. The radiosonde temperature measurement from the altitude closest to each DTS measurement altitude was used for comparison. The 23:30 UTC radiosonde and DTS temperature measurements between the surface and the maximum altitude of the fiber (839 m AGL) showed a correlation of $R^2$ of 0.99 and an RMSE of 0.6 °C.





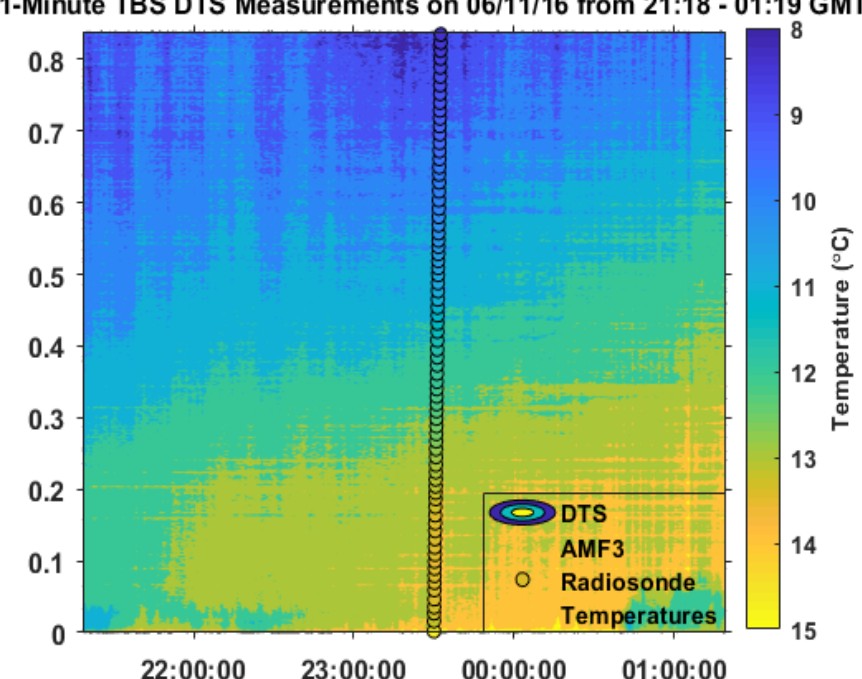

**Figure 13:** TBS DTS measurements collected once per minute with concurrent free balloon radiosonde launch at 23:30 UTC.

### 3.2.2 Comparison of DTS temperature measurements with TBS iMet radiosonde temperatures at the same altitude

While the comparison of TBS DTS measurements and free flight radiosonde measurements is informative, the dataset is limited

5    since only two radiosondes are launched daily from the AMF3. In order to compare a larger number of samples (197 samples over a 20 °C range were available), temperatures from an iMet radiosonde suspended on the TBS tether were compared with DTS measurements collected at the same altitude over nine TBS flights from 2016 - 2017. DTS 1 m spatial resolution and 60 second temporal data were averaged over 10 m to match the simultaneous AMF3 radiosonde vertical resolution. During two of these nine flights two channels were used on the DTS. While the multimode fiber used for TBS DTS measurements is

10    white, some excess heating due to solar radiation could still occur. In an effort to assess and correct for this excess heating, the linear fits between proprietarily solar radiation-corrected iMet radiosonde temperatures and DTS temperatures for each flight were applied to the DTS temperature values. The mean RMSE between the iMet radiosonde temperatures and uncorrected DTS temperatures was 0.39 °C, improving to 0.32 °C after the radiation-correction factors were applied to DTS temperatures.

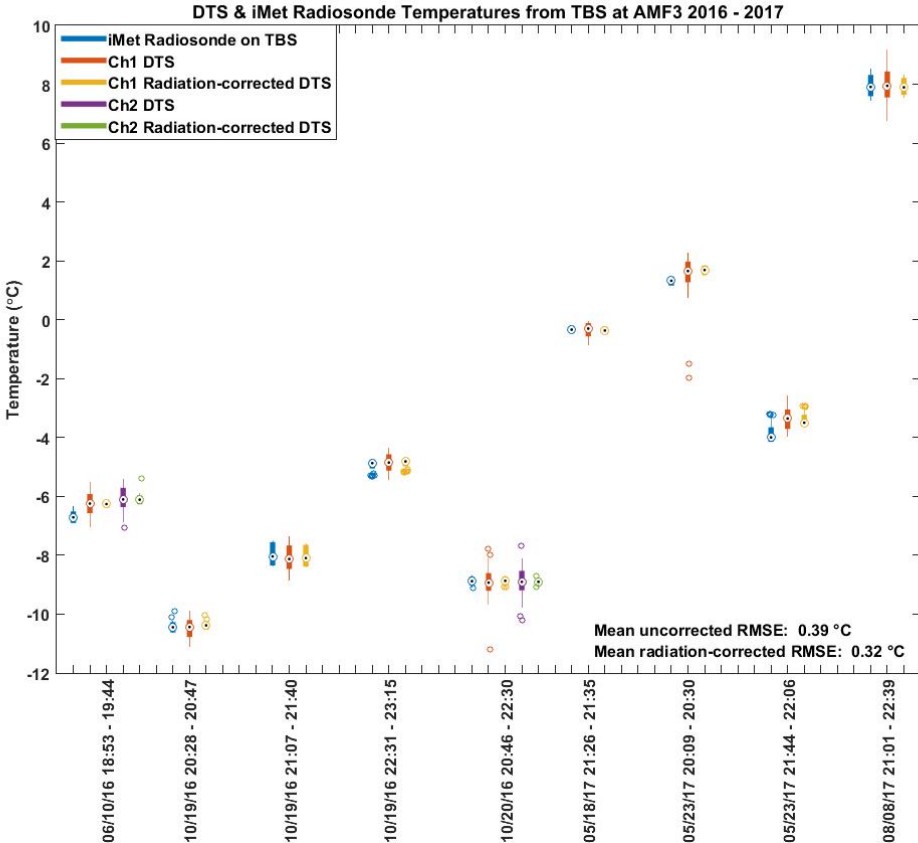

**Figure 14:** DTS and iMet radiosonde temperatures at matching altitudes from nine TBS flights at the AMF3 from 2016 – 2017.

### 3.2.3 7/10/18 and 7/11/18 DTS measurements with POPS aerosol instruments

During the POPEYE field campaign a Silixa XT DTS system was operated on the TBS using 50 micron multimode optical

fiber suspended along the tether. Temperature measurements were collected every 30 – 60 s with a spatial resolution of 0.65

cm.  Two POPS (Printed Optical Particle Spectrometers) were suspended along the tether. One POPS was operated just below

the balloon in order to reach the maximum possible altitude, which was ideally above cloud top. A second POPS was generally

operated lower on the tether near cloud base.

On 7/10/18, the continuous DTS temperature profiles and iMet radiosonde temperatures reveal a cool layer at the surface

below 100 m with a 3-4 °C warmer layer between 150 and 800 m, then another cooler layer above the inversion from 800 m

to 1 km. The AMF3 radiosonde launch at 23:30 measured a similar temperature profile. The particle concentration per second

measured by the POPS demonstrates increased particle concentration within the temperature inversion, with fewer particles

above the inversion and in the surface-cooled layer. The surface layer warmed in the afternoon and the height of the inversion

layer increased with time.  On 7/11/18 the surface layer below 200 m was roughly 2 °C cooler than on the previous day, as

were temperatures in the inversion layer between 200 m and 1.2 km. POPS particle concentration per second within the





inversion was approximately double that observed on the previous day. The height of the inversion layer decreased between 18:30 and 19:30, and a shallow ~50 m-deep inversion layer was isolated around 400 m after 19:30 with a 200 m-deep cooler layer above. An iMet radiosonde on the tether corroborated this shallow inversion layer measured by the DTS temperature profiles. POPS particle concentrations were elevated within this shallow warm layer and exhibited increased variability.

5   Similar to the day prior, POPS particle concentrations decreased above and below the inversion.

**Figure 15:** 7/10/18 – 7/11/18 TBS DTS profiles at AMF3 with TBS iMet temperatures (squares), free-flight radiosonde temperatures (diamonds), and POPS particle concentrations per second (circles).





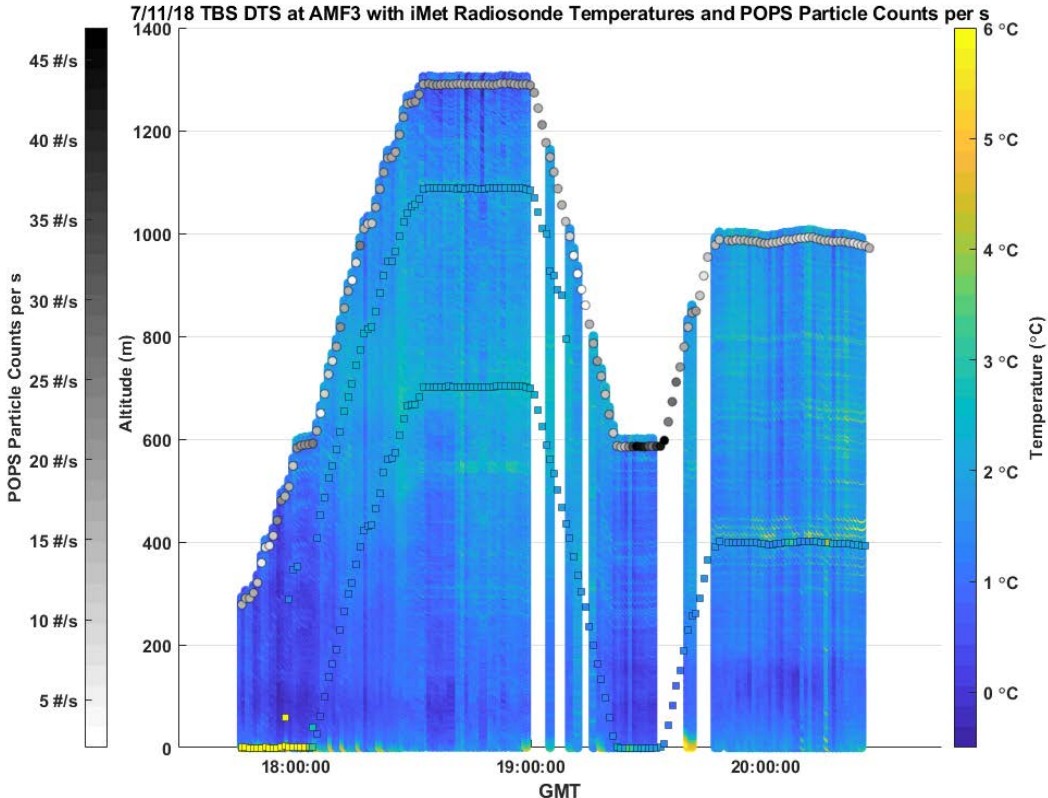

**Figure 16:** 7/11/18 TBS DTS profiles at AMF3 with TBS iMet temperatures (squares) and POPS particle concentrations per second (circles).

**3.2.4 DTS temperature calibration source and fiber optic rotary joint impacts on measurement accuracy**

The DTS collected over 300 hours of 30s measurements with two fibers during the POPEYE field campaign. One fiber did not include a rotary joint and was in use only when the balloon was not ascending or descending. The other fiber was installed with a fiber optic rotary joint (FORJ) and measured continuously. The DTS measurements were calibrated with a reference temperature sensor installed on the tether at the maximum-altitude ends of the fibers. An iMet radiosonde and iMet XQ2 sensor were both used to provide reference temperatures. The DTS temperatures were averaged vertically over 5 m in order to

compare with temperatures from simultaneous radiosonde profiles. The average correlation coefficients and RMSEs between the DTS fiber measurements calibrated with the iMet radiosonde or XQ2 sensor, collected with or without a FORJ, and free-flight radiosonde temperatures are shown in Table 2.

The iMet radiosonde and XQ2 sensors performed almost identically as reference temperature sources. The FORJ and non-FORJ temperatures correlated to each other at 0.74 and had an RMSE of 0.5 °C. Both the FORJ and non-FORJ measurements

correlated to radiosondes at 0.97 with RMSEs from 0.4 – 0.6 °C.





**Table 2.** Correlation coefficients and RMSEs for iMet-1-RSB radiosonde or iMet XQ2-calibrated DTS data, and DTS data collected with or without a Fiber Optic Rotary Joint (FORJ).

|  | Correlation | RMSE (°C) |
|---|---|---|
| **Mean iMet-Calibrated, XQ-2-Calibrated** | 0.76 | 0.49 |
| **Mean iMet-Calibrated FORJ, iMet-Calibrated non-FORJ** | 0.74 | 0.51 |
| **Mean XQ2-Calibrated FORJ, XQ2-Calibrated non-FORJ** | 0.74 | 0.50 |
| **FORJ iMet-Calibrated vs Radiosonde** | 0.97 | 0.49 |
| **FORJ XQ2-Calibrated vs Radiosonde** | 0.97 | 0.60 |
| **Non-FORJ iMet-Calibrated vs Radiosonde** | 0.97 | 0.43 |
| **Non-FORJ XQ2-Calibrated vs Radiosonde** | 0.97 | 0.46 |

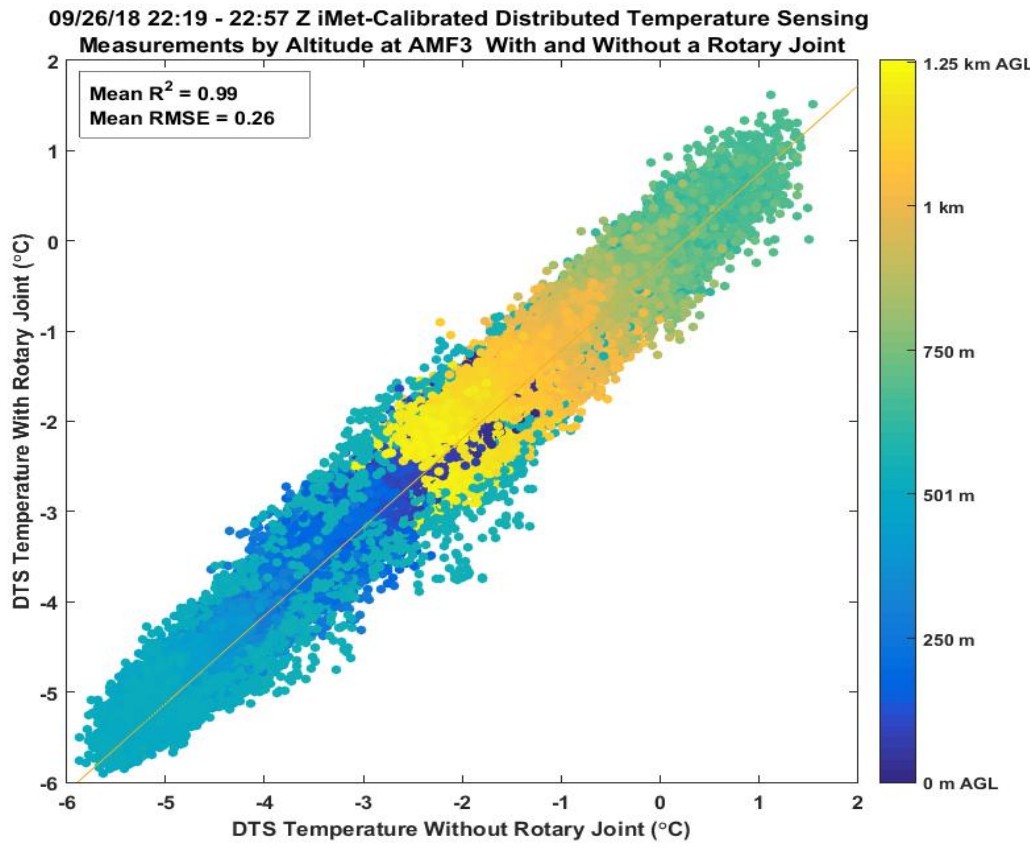

**Figure 17:** 9/26/18 TBS DTS measurements with and without a fiber optic rotary joint (FORJ).



## 4 Conclusions

In situ, vibrating wire-based measurements of supercooled liquid water within Arctic clouds collected using a tethered balloon system have been evaluated against surface-based remote sensing and radiosonde-derived measurements. First, a free-balloon sounding that occurred during the TBS flight was analyzed using the enthalpy equation for cloud water (LWC) in the commercial software RAOB. The supercooled liquid water contents calculated by RAOB from the free radiosonde flight at the altitudes of two vibrating-wire Anasphere SLWC sondes on the tethered balloon were both 0.14 $g/m^3$. The maximum SLWC observed by the highest altitude SLWC sonde at the time of the radiosonde flight was 0.14 $g/m^3$, while the maximum SLWC observed by the lower altitude sonde was 0.05 $g/m^3$. While the absolute uncertainty between the two measurements is difficult to determine given the standard deviation of 120 m in the cloud base measured by the ceilometer at the time of the radiosonde flight, and the spatial variation between the TBS vibrating-wire sonde and AMF3 radiosonde measurement sites, the TBS Anasphere SLWC sondes and RAOB LWC measurements agree within 0.1 $g/m^3$.

Second, SLWC values from in situ TBS Anasphere sondes were compared with adiabatically-distributed LWP values from the AMF3 MWR for two TBS flights in October 2016. During the first flight the cloud base varied significantly. Both SLW measurements closely identified the occurrence of SLW in time, although the magnitude differed, resulting in an $R^2$ value of 0.38. Again, given the uncertainties in the cloud base height, the altitude of the vibrating-wire sonde, and the relative positions of each, the temporal agreement of SLWC detection between the two sensors is significant. For the second flight the $R^2$ value was 0.79, with MWR SLWC values averaging 0.03 $g/m^3$ higher. The low bias of the TBS SLWC sonde can be partially attributed to the vibrating wire experiencing maximum ice loading and incompletely shedding multiple times during the flight, resulting in slightly low SLWC values, due to the failure of the wire to completely return to its un-iced initial frequency.

The Anasphere sensor was deployed alongside an alternative SLWC sensor developed at the University of Reading in order to compare data sensed using different available vibrating wire-based sensors. There was general good agreement between the two vibrating sensors and also between them and the MWR measurements (LWP ±0.008 mm for the Reading sensor and ±0.003 mm for the Anasphere sensor during ascents and 0.1 $g/m^3$ RMSE between the two sensors for the flight). Some potential for additional variation related to ice load saturation and instrument geometry may be important and should be considered when comparing data collected using different sensors.

Mean values of SLWC from the Anasphere sondes deployed on the TBS for 43 flights were compared with mean SLWC values calculated from AMF3 radiosonde soundings that occurred during each TBS flight using the enthalpy lapse-rate equation for cloud water (LWC) in the commercial software RAOB. The mean SLWC values measured by the in situ vibrating wire sondes averaged 0.045 $g/m^3$ higher than the mean SLWC values calculated from the radiosonde flights. SLWC values from vibrating sondes were < 0.05 $g/m^3$ for 38% of the samples, which is close to previous aircraft measurements in supercooled stratiform clouds, < 0.05 $g/m^3$ for 36% of the samples (e.g., Sand et al., 1984).



Recurring TBS Anasphere SLWC sonde deployments occurred at the ARM AMF3 during fall and spring months between 2015 and 2017. The largest SLWC values were measured during May and June, with smaller values being measured in fall and early spring. Larger SLWC values were measured above 400 m altitude and at temperatures between -14 °C and -2 °C. DTS measurements collected between the surface and the balloon were compared with concurrent radiosonde temperature

measurements. The effect of different calibration measurement source instruments upon the DTS measurement accuracy were evaluated, as was the use of a fiber optic rotary joint. Radiosonde and DTS temperature measurements between the surface and the maximum altitude of the fiber, 0.84 km AGL, correlated with an $R^2$ of 0.99 and an RMSE of 0.6 °C. In order to compare a larger number of samples (197 samples over a 20 °C range), temperatures from an iMet radiosonde suspended on the TBS tether were compared with DTS measurements collected at the same altitude over nine TBS flights from 2016 – 2017.

The mean RMSE between the iMet radiosonde temperatures and uncorrected DTS temperatures was 0.39 °C, improving to 0.32 °C after solar radiation-correction factors were applied to DTS temperatures.

DTS temperature measurements collected with and without a fiber optic rotary joint correlated to each other at 0.74 and had an RMSE of 0.5 °C. FORJ DTS measurements had an average correlation of 0.97 with radiosonde temperatures, and RMSE values of 0.5 and 0.6 °C for the iMet-1-RSB radiosonde and iMet XQ2-calibrated DTS datasets, respectively. The non-FORJ

DTS measurements had an identical average correlation of 0.97 with radiosonde temperatures, and RMSE values of 0.43 and 0.46 °C for the iMet-1-RSB and iMet XQ2-calibrated DTS datasets, respectively.

The similar RMSE values between the DTS and radiosonde datasets both with and without an installed FORJ indicates the presence of the FORJ has very limited impact on the measurement accuracy of DTS measurements. For context, the stated temperature measurement accuracies are +/- 0.2 °C, +/- 0.3 °C, and +/- 0.2 °C, for the iMet-1-RSB radiosonde, iMet XQ2, and

Vaisala RS-92 sensors, respectively. The average RMSE of 0.4 – 0.6 °C between DTS temperature and radiosonde temperature measurements indicates DTS measurements from TBSs provide accurate, highly spatially and temporally-resolved, persistent temperature profiles within the lowest two kilometers of the atmosphere. The approximate average RMSE of 0.5 °C between the DTS measurements and Vaisala radiosonde measurements may partially be attributed to spatial disparity between the tethered and free-flight sensors, noting that free-flight radiosonde launches were intentionally launched upwind of the tethered

balloon during simultaneous flights to avoid potential entanglement. Additional differences in temperature measurement may be due to measurement bias between the iMet-1-RSB radiosondes and iMet XQ2 sensors used to calibrate the TBS DTS measurements, and the Vaisala RS-92 sondes used in AMF3 radiosonde launches. Some RMSE may also be attributed to uncertainty in the GPS-reported altitude from each sensor, given the stated vertical accuracy of +/- 12m of the iMet XQ2 sensor, +/- 15m of the iMet-1-RSB, and +/- 20 m of the Vaisala RS-92. The relationship between iMet sensor temperatures

and Vaisala radiosonde temperatures will be investigated in future TBS flights.

**Acknowledgements**

We acknowledge the US Department of Energy Atmospheric Radiation Measurement program and Sandia National Laboratories for logistics support. This work was supported in part by the Laboratory Directed Research and Development





program at Sandia National Laboratories, a multi mission laboratory managed and operated by National Technology and Engineering Solutions of Sandia, LLC., a wholly owned subsidiary of Honeywell International Inc., for the U.S. Department of Energy's National Nuclear Security Administration under contract DE-NA0003525. MA, KN, RGH, PDW, and GM acknowledge support from NERC Grant No. NE/P003362/1 (VOLCLAB) for development of the Reading SLWC sensor. KN

acknowledges support through NERC Independent Research Fellowships NE/L011514/1 and NE/L011514/2.  SK was supported by the Deutsche Forschungsgemeinschaft (DFG, German Research Foundation) under grant KN 1112/2-1 as part of the Emmy-Noether Group OPTIMIce.

The authors wish to thank Dr. Gijs de Boer for leading the ICARUS and POPEYE field campaigns.  John Shewchuk of Eosonde Research Services provided documentation regarding RAOB software calculations.  Dr. John Bognar of Anasphere, aided in

the development of the SLWC Anasphere sonde data processing methodology.  Dr. David Serke of UCAR provided guidance regarding the collection of SLWC data in flight.  Allen Jordan and Emrys Hall of NOAA developed the Skysonde software for iMet radiosondes that was used to collect all iMet radiosonde-derived data used in this research.  Dr. Scott Tyler of University of Nevada, Reno and CTEMPS provided initial training on the use of DTS and DTS data processing.

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
