# Peer review of "Evaluation of ARM Tethered Balloon System instrumentation for supercooled liquid water and distributed temperature sensing in mixed-phase Arctic clouds"

_Atmospheric Measurement Techniques, 2019_

## Referee Comment (RC1) · Anonymous Referee #2 · 28 May 2019

Review of

**Evaluation of ARM Tethered Balloon System instrumentation for supercooled liquid water and distributed temperature sensing in mixed-phase Arctic clouds**

by Darielle Dexheimer et al.

General Comments:

The manuscript describes an evaluation of tethered aerostats helikites instrumented for cloud measurements. This reviewer has several problems with the way this manuscript is presented, and how the measurements are evaluated.

1. The manuscript approaches the subject as if its TBS is a novel sampling technique that has never before been attempted, when in fact, the literature contains references to cloud sampling with tethered balloons for nearly the past 40 years. Kitchen and McClatchey (1981) flew an instrumented aerostat in cumulus clouds in the UK. Several investigators have followed suit, including measurements in the Arctic (Lawson et al. 2011) and at the South Pole (Lawson and Gettelman 2014). The manuscript needs to provide context for their research by discussing the previous efforts and explain why their current approach is an improvement, or how it differs.

2. The methodology used to process and present the measurements does not do justice to the larger variance in the different techniques being compared. While the variance can be seen in some of the measurements (e.g. time series in Fig. 3; scatterplots in Fig. 7), the overall message presented is that there is relatively good agreement in the measurements and that therefore they are useful. In fact, it is not at all clear that the vibrating wire technique is of any value in quantitatively measuring SLWC. The manuscript compares mean values of SLWC measured from the two vibrating wire techniques to validate the instruments when the variance in the measurements is larger than the mean; this is not an acceptable method for validating instrument performance. I agree that the technique may be useful as an SLWC detection method, and perhaps it can differentiate between low values i.e., < a few tenths g m$^{-3}$ and high values (> ~ 1 g m$^{-3}$), but even this cannot be validated since high SLWC measurements are not presented.

3. I do feel that the DTS measurements are sufficiently accurate to be useful, but the validation is against a radiosonde. If the comparison is good, why not just use radiosonde data? Certainly it is more cost effective than the complexity involved with launching a tethered aerostat or helikite. One possible use of the DTS measurements is to better define low-level inversions, but I did not find a meteorological use for this presented in the paper.

4. Before the manuscript is fit for publication it should be modified considerably. The measurements are presented in a manner that embellishes their actual efficacy. The manuscript should present the data in an biased manner and focus on the ability of the TBS to make routine, long-term measurements, but eliminate the impression that the instruments themselves are useful for making quantitative measurements.

Specific Comments:

Introduction: Add TBS background references and discuss.

Page 3, Lines 11 – 14: This is incorrect. There are much larger helikytes currently in service. For example, Bodenschatz, in Goettingen has deployed a 250 m$^3$ helikyte from a ship in the Arctic.

Figure 3: The figure and text states that the difference in mean values of SLWC between the two Anasphere devices is 0.01, 0.02 and 0.06 g m$^{-3}$. This is meaningless without also giving some measurement of the variance, standard deviation or show a scatterplot. Even two noise signals can have the same mean. The figure suggests that there is little correlation between the two devices.

Figure 7: I applaud the authors for showing the scatterplots in this figure, but the interpretation of the results does not fit the data. I am not sure how an R$^2$ value is even relevant. The scatterplot and time series tell the story. There is virtually no correlation in the measurements. The best that one can ascertain here is that the TBS is an SLWC detector.

Figures 8 and 9: This is a good representation of the measurements, but again, the interpretation is off the mark. The statement on p. 16,

"Overall, the two sensors seem to provide broadly similar SLWC profiles, but not without some discrepancies. This increases confidence in the use of each of them and provides independent verification of the measurements."

grossly overstates the efficacy of the measurements. The profiles in Figure 9 often diverge and differ by factors of 2 to 3. The authors need to show measurements with better correlation over a much larger range of SLWC's, and the sensors need to agree with SLWCad to convince readers that the instruments can be used quantitatively. Several aircraft campaigns have shown that single-layer ice-free Arctic stratus clouds typically have adiabatic SLWC profiles up to 60 to 90% of the distance between cloud base and cloud top. Of course, this will be challenging if the balloons cannot achieve higher altitudes.

Page 17, Lines 4 – 5: Equation 3 computes the adiabatic maximum SLWC. Since the vibrating wires measured a higher value than adiabatic, the measurements exceed the theoretical maximum. Also, once again, only reporting means does not show the variance in the measurements, which should also be reported, preferably by showing a scatterplot.

Page 17, Line 8: I have examined the Sand et al. (1984) article and I cannot find any text or figures that support the claim in the manuscript. Please show how 36% of the samples < 0.05 g m$^{-3}$ was derived from the Sand paper. Also, the minimum measurement sensitivity of the J-W used in the Sand study was 0.1 g m$^{-3}$ due to drift and noise affecting the baseline. The FSSP was used for lower values, but its fundamental measurement is drop size and raising drop size to the third power increases sizing errors commensurately. Thus, the FSSP is not a fundamental measurement of LWC. Reference to the Sand paper is a stretch and appears to be an unsubstantiated attempt to validate the vibrating wire measurements.

Conclusions: The conclusions are very misleading. The measurements are limited to very low values of SLWC, so the dynamic range over which the measurements are constrained is tiny. Means are compared within this very limited dynamic range, making the comparison look good (e.g., the means of the measurements compared within 0.1 g m$^{-3}$). This a totally unacceptable methodology for presenting the measurements. Showing the time series comparisons and scatterplots is representative and these figures are the saving grace of this manuscript. The paper should focus on the mechanics of how the TBS systems are operated, and that they can be successfully operated for extended periods of time. The vibrating wire technology is used on commercial aircraft to detect icing conditions, not to quantify icing rate, even though there was an attempt in the 80's to do so (i.e., Baumgardner and Rodi 1989). While Baumgardner and Rodi reported that independent laboratory calibration of the Rosemount icing probe improved its performance, they also reported that each of the four instruments tested had different mass sensitivities. There has been virtually no quantitative reporting of SLWC from an airborne Rosemount Icing probe in the literature. In time perhaps better SLWC measurement techniques applicable to TBS will be developed, implemented and reported.

**References:**

Baumgardner, D. and A. Rodi, 1989: Laboratory and Wind Tunnel Evaluations of the Rosemount Icing Detector. *J. Atmos. Oceanic Technol.,* **6**, 971–979.

Kitchen, M. and S. J. Caughey, 1981: Tethered-balloon observations of the structure of small cumulus clouds. *Q.J.R. Meteorol. Soc*., **107**, 853-874.

Lawson, R. P., K. Stamnes, J. Stamnes, P. Zmarzly, J. Koskuliks, C. Roden, Q. Mo, M. Carrithers, 2011: Deployment of a Tethered Balloon System for Cloud Microphysics and Radiative Measurements at Ny-Ålesund and South Pole, *J. Atmos. Oceanic Technol.* **28**, 656 – 670.

Lawson, R. P. and A. Gettelman. 2014: Impact of Antarctic mixed-phase clouds on climate. *Proc. Nat. Acad. Sci*., **111**, 18156–18161.

---

## Referee Comment (RC2) · Anonymous Referee #1 · 29 May 2019

Dexheimer et al. presented a well-written manuscript about cloud microphysical measurements in a sparsely sampled region. The usage of a tethered balloon or balloon-kite is well motivated as airplanes cannot do this type of measurement in supercooled liquid water clouds due to icing. The authors have shown that using a distributed temperature sensing unit aboard a tethered balloon can provide vertical temperature profiles that are about as accurate as a radiosonde. Disparities in the measured supercooled liquid water content were discussed adequately. In general, this is a good manuscript that is definitely worth to be published. However, some clarification is

needed in some places, and some (mostly technical) things should be fixed prior to publication.

I have only a few general comments for further improvement of the manuscript:

1) There are many nice figures in the paper, but not all of them are actually referenced in the paragraph describing them. A summary of the missing figure references is given in the detailed discussion.

2) The role of wind speed for the measurements should be discussed. Do you have measurements of wind speed on the TBS? It might be very helpful to see if some of the observed disparities are more common under specific wind conditions, and if there might be an influence due to the presence of the balloon in very low winds (e.g. seeding and cloud glaciation due to Wegener-Bergeron-Findeisen) or issues in the SLWC measurement due to strongly varying winds. If I interpret Hill (1994) correctly, constant but not too high wind speeds are preferred for the SLWC sondes.

Detailed comments:

Page 1, Line 14: Please provide a bit more clarification of which kind of in situ atmospheric measurements you did.

Page 6, Figure 2: The beginning of the second sentence in the caption seems to be redundant. If I understood it correctly, the Anasphere SLWC sonde next to the InterMet radiosonde is shown on the left, the Anasphere SLWC sonde above the Anasphere tethersonde is shown in the middle and the Reading SLWC sonde is shown on the right. I would ask you to reformulate the caption accordingly.

Page 7, Figure 3: Figure 3 provides a good qualitative comparison of the sonde pairs, but I would recommend to include an additional figure to compare the measured distribution functions of SLWC. For that I would use only those times where both sondes had valid, non-zero readings, calculate the cumulative distribution function (or probability distribution function, whatever you like better) for each sonde and put the CDFs /

PDFs for each pair of sondes in one plot.

Page 9, Line 10: I am pretty sure that you mean Fig. 4 instead of Fig. 7 for the figure reference.

Page 11: I do not see a figure reference to Fig. 6.

Page 12, Figure 6: There is almost no contrast in the data points showing the SLWC. I would suggest plotting the values below the detection limit either white or transparent to enhance the contrast for the data that actually matter.

Page 14: I do not see a figure reference to Fig. 8.

Page 17: I do not see a figure reference to Fig. 10, Fig. 11 and Fig. 12.

Page 19: I do not see a figure reference to Fig. 13.

Page 20: I do not see a figure reference to Fig. 14.

Page 20, Line 14: Please describe the radiation correction of the DTS in more detail (reference or equation, if possible). It might be best to explain the radiation correction in Subsection 2.3.

Page 21, Line 10: When looking closely at Fig. 15, I do not see an increase of the temperature with altitude by more than 1 to 1.5 degrees Celsius. Please clarify where exactly the 3-4 °C warmer layer is located. Could it be the wrong figure that is put in the manuscript?

Page 21: I do not see a figure reference to Fig. 15 and Fig. 16.

Page 22, Fig. 15 and 16: It is hard to actually see where the inversion is located in the vertical temperature profile. Would it be more intuitive to localize if you plot the potential temperature instead of the temperature? And is there an explanation for the high near-surface temperatures in Fig. 16 around 19:40 UTC?

Page 23: I do not see a figure reference to Fig. 17.

---

## Author Comment (AC1) · 11 Jul 2019

We thank all of the reviewers for their valuable comments. Below, we have added some responses to the comments submitted.

Reviewer 1 General Comments: 1) The manuscript approaches the subject as if its TBS is a novel sampling technique that has never before been attempted, when in fact, the literature contains references to cloud sampling with tethered balloons for nearly the past 40 years. Kitchen and McClatchey (1981) flew an instrumented aerostat in

[Figure]

cumulus clouds in the UK. Several investigators have followed suit, including measurements in the Arctic (Lawson et al. 2011) and at the South Pole (Lawson and Gettelman 2014). The manuscript needs to provide context for their research by discussing the previous efforts and explain why their current approach is an improvement, or how it differs.

Response 1) We thank the reviewer for their comments, but believe historical TBS flight activities are discussed significantly in the existing manuscript on pg. 2, lines 16-26. We did add a reference to the highly relevant work discussed in Lawson 2011 and have included a summary of that publication to the existing discussion on pg. 2. We agree that we should better explain why our current approach is an improvement or how it differs. Towards that end we have revised the following text to read: "The work discussed herein pertains to the new capability of using tethered balloon systems within restricted airspace for persistent, repeatable, interannual flights inside Arctic mixed phase clouds, with supercooled liquid water sondes and distributed temperature sensing optical fiber systems."

2) The methodology used to process and present the measurements does not do justice to the larger variance in the different techniques being compared. While the variance can be seen in some of the measurements (e.g. time series in Fig. 3; scatterplots in Fig. 7), the overall message presented is that there is relatively good agreement in the measurements and that therefore they are useful. In fact, it is not at all clear that the vibrating wire technique is of any value in quantitatively measuring SLWC. The manuscript compares mean values of SLWC measured from the two vibrating wire techniques to validate the instruments when the variance in the measurements is larger than the mean; this is not an acceptable method for validating instrument performance. I agree that the technique may be useful as an SLWC detection method, and perhaps it can differentiate between low values i.e., < a few tenths g m-3 and high values (> ∼ 1 g m-3), but even this cannot be validated since high SLWC measurements are not presented.

Response 2) Again we thank the reviewer, but believe the discrepancies in the vibrating wire measurements are thoroughly discussed in the manuscript. In order to allow the reader to judge the efficacy of the vibrating wire SLWC measurements, comparisons between the SLWC measured by more than one vibrating wire instrument are shown, in addition to how the vibrating wire-measured SLWC compares with radiosonde-derived SLWC and MWR-derived SLWC. We admit we are unclear what the viewer is referring to by "The manuscript compares mean values of SLWC measured from the two vibrating wire techniques to validate the instruments . . . " since no comparison of mean values of SLWC measured from the two vibrating wire techniques as a validation of the instruments is presented. The mean differences between the pairs of Anasphere sondes in Figure 3 is presented to depict the noise or relative uncertainty in the Anasphere sonde measurements. If the reviewer is referring to Figure 9, which depicts LWP measured by the two different vibrating wire sondes and the MWR, the closest relationship between the mean and variance of the SLWC values occurs for the Anasphere sonde during the first descent with values of 0.02 and 0.0007 g/m3 for the mean and variance, respectively.

3) I do feel that the DTS measurements are sufficiently accurate to be useful, but the validation is against a radiosonde. If the comparison is good, why not just use radiosonde data? Certainly it is more cost effective than the complexity involved with launching a tethered aerostat or helikite. One possible use of the DTS measurements is to better define low-level inversions, but I did not find a meteorological use for this presented in the paper.

Response 3) This is a good suggestion and the authors have added the following text to the Introduction in regard to the advantages of using tethered balloon DTS measurements compared to radiosonde data. "DTS provides greatly improved spatial and temporal resolution of temperature compared to radiosonde measurements and allows measurements to be collected continuously between the balloon and the surface for the duration of the tethered balloon flight."

4) Before the manuscript is fit for publication it should be modified considerably. The measurements are presented in a manner that embellishes their actual efficacy. The manuscript should present the data in an biased manner and focus on the ability of the TBS to make routine, long-term measurements, but eliminate the impression that the instruments themselves are useful for making quantitative measurements.

Response 4) We thank the reviewer for their comments but believe the ability of the TBS to make routine, long-term measurements is discussed significantly in the Introduction and sections 2.1.2, 3.1.4, 3.1.5, 3.2.2, and 3.2.4 of the manuscript. We also believe the comparisons of the SLWC vibrating wire sonde data between both sondes, the MWR, and simultaneous radiosonde data are fairly presented and the discrepancies associated with the vibrating wire sonde measurements are extensively discussed in sections 3.1.1 – 3.1.3.

Reviewer 1 Specific Comments:

1) Introduction: Add TBS background references and discuss.

Response 1) Please see response to Reviewer 1 General Comments.

2) Page 3, Lines 11 – 14: This is incorrect. There are much larger helikytes currently in service. For example, Bodenschatz, in Goettingen has deployed a 250 m3 helikyte from a ship in the Arctic.

Response 2) We are uncertain what the reviewer is referring to in this case, since pg 3, lines 11-14 of the manuscript discuss the MWR. Perhaps the reviewer is referring to pg 3, line 31 – 32: "Helikites are typically used for flights with desired altitudes to 700 m Above Ground Level (AGL),a maximum payload of less than 10 kg, and in surface wind speeds less than 11 m/s." For clarity the authors have revised this line to begin with "In the ARM TBS, helikites . . . "

3) Figure 3: The figure and text states that the difference in mean values of SLWC between the two Anasphere devices is 0.01, 0.02 and 0.06 g m-3. This is meaningless

without also giving some measurement of the variance, standard deviation or show a scatterplot. Even two noise signals can have the same mean. The figure suggests that there is little correlation between the two devices.

Response 3) We thank the reviewer and have added pdfs of each sonde's measurements as suggested by Reviewer #2 to better illustrate the data collected by the two devices.

4) Figure 7: I applaud the authors for showing the scatterplots in this figure, but the interpretation of the results does not fit the data. I am not sure how an R2 value is even relevant. The scatterplot and time series tell the story. There is virtually no correlation in the measurements. The best that one can ascertain here is that the TBS is an SLWC detector.

Response 4) As discussed in pg 12, line 18 through pg 13, line 9 the authors view the timing of SLWC detection between the airborne SLWC sensor and the surface-based MWR as significant. This result is especially relevant considering the uncertainties associated with the ceilometer-reported cloud base, the GPS-reported altitude of the airborne SLWC sonde in relation to the cloud base, and the determination of the cloud top altitude from the KAZR data, in addition to the variation in the cloud base on 10/15/16, and the saturation of the vibrating wire on the sonde and resulting ice shedding events on 10/20/16. As the reviewer mentions, the authors have included scatter plots of the results so the reader can evaluate the relationship between the two measurements.

5) Figures 8 and 9: This is a good representation of the measurements, but again, the interpretation is off the mark. The statement on p. 16, "Overall, the two sensors seem to provide broadly similar SLWC profiles, but not without some discrepancies. This increases confidence in the use of each of them and provides independent verification of the measurements." grossly overstates the efficacy of the measurements. The profiles in Figure 9 often diverge and differ by factors of 2 to 3. The authors need to show measurements with better correlation over a much larger range of SLWC's,

and the sensors need to agree with SLWCad to convince readers that the instruments can be used quantitatively. Several aircraft campaigns have shown that single-layer ice-free Arctic stratus clouds typically have adiabatic SLWC profiles up to 60 to 90% of the distance between cloud base and cloud top. Of course, this will be challenging if the balloons cannot achieve higher altitudes.

Response 5) The reviewer is correct in that there are differences in the magnitude of the SLWC detected by the various sensors, but this is to be expected for some of the reasons detailed on pg 16, lines 5-12. To this we have also added "The ice accumulation on the vibrating wire of each sonde may be dependent upon the upwind or downwind orientation of the sonde with respect to the balloon tether". The measurement of SLWC is a difficult one to make and these sensors are designed to be as inexpensive and lightweight as possible so as to be carried on disposable balloons, therefore the fact that the sensors roughly track each other and are generally within a factor of 2 is a reasonable achievement. If this was a comparison between aircraft instruments then we would expect better agreement, but one should not expect such accuracy from inexpensive disposable instruments.

6) Page 17, Lines 4 – 5: Equation 3 computes the adiabatic maximum SLWC. Since the vibrating wires measured a higher value than adiabatic, the measurements exceed the theoretical maximum. Also, once again, only reporting means does not show the variance in the measurements, which should also be reported, preferably by showing a scatterplot.

Response 6) The theoretical calculation of adiabatic SLWC results in mean SLWC values of < 0.05 g/m3 for 91% of 43 in-cloud TBS flights, which is inconsistent with in situ SLWC measurements within mixed-phase Arctic clouds from the literature. The in situ measurements collected by the TBS SLWC sondes exhibit better agreement with the literature values. This result depicts the value of the TBS to collect in situ atmospheric measurements and better refine cloud parameterization schemes used in climate models. A pdf and cdf of the SLWC data has also been added and the

text updated to read "The probability density and cumulative distribution functions of all SLWC data collected with the Anasphere sondes on the TBS for the 43 flights are shown in Figure 12, and the probability density of SLWC values < 0.05 g/m3 was 34%."

7) Page 17, Line 8: I have examined the Sand et al. (1984) article and I cannot find any text or figures that support the claim in the manuscript. Please show how 36% of the samples < 0.05 g m-3 was derived from the Sand paper. Also, the minimum measurement sensitivity of the J-W used in the Sand study was 0.1 g m-3 due to drift and noise affecting the baseline. The FSSP was used for lower values, but its fundamental measurement is drop size and raising drop size to the third power increases sizing errors commensurately. Thus, the FSSP is not a fundamental measurement of LWC. Reference to the Sand paper is a stretch and appears to be an unsubstantiated attempt to validate the vibrating wire measurements.

Response 7) We thank the reviewer for this close review. On closer inspection the results referred to were in fact modified from Sand et al. (1984) and are reported at www-das.uwyon.edu/∼geerts/cwx/notes/chap08/moist_cloud.html. We have updated the text and references to refer to the University of Wyoming reference.

8) Conclusions: The conclusions are very misleading. The measurements are limited to very low values of SLWC, so the dynamic range over which the measurements are constrained is tiny. Means are compared within this very limited dynamic range, making the comparison look good (e.g., the means of the measurements compared within 0.1 g m-3). This a totally unacceptable methodology for presenting the measurements. Showing the time series comparisons and scatterplots is representative and these figures are the saving grace of this manuscript. The paper should focus on the mechanics of how the TBS systems are operated, and that they can be successfully operated for extended periods of time. The vibrating wire technology is used on commercial aircraft to detect icing conditions, not to quantify icing rate, even though there was an attempt in the 80's to do so (i.e., Baumgardner and Rodi 1989). While Baumgardner and Rodi reported that independent laboratory calibration of the Rosemount icing probe

improved its performance, they also reported that each of the four instruments tested had different mass sensitivities. There has been virtually no quantitative reporting of SLWC from an airborne Rosemount Icing probe in the literature. In time perhaps better SLWC measurement techniques applicable to TBS will be developed, implemented and reported.

Response 8) As we have added the cdf and pdf of all SLWC values at the good suggestion of the reviewer, the reader can now observe that 40% of the SLWC data collected were >= 0.1 g/m3. The text of the Conclusion has also been updated to refer to the SLWC data in the pdf/cdf figure.

Please also note the supplement to this comment:
https://www.atmos-meas-tech-discuss.net/amt-2019-117/amt-2019-117-AC1-supplement.pdf

[Figure]

SLWC Values from TBS Anasphere Sondes from 2016 - 2017

Probability Density

SLWC (g/m³)

SLWC Values from TBS Anasphere Sondes from 2016 - 2017

Cumulative probability

— TBS Anasphere Sonde SLWC
— Weibull Distribution

SLWC (g/m³)

**Fig. 1.** Probability density and cumulative distribution functions of all SLWC data collected with the Anasphere sondes on the TBS for the 43 flights.

**10/17/16 PDF of Anasphere SLWC Values**

- Anasphere SLWC V8
- Anasphere SLWC #3

Probability Density / SLWC (g/m$^3$)

**11/16/16 PDF of Free-Flight Anasphere SLWC Values**

- Anasphere SLWC #6
- Anasphere SLWC #2

Probability Density / SLWC (g/m$^3$)

**04/03/17 PDF of Anasphere SLWC Values**

- Anasphere SLWC #8
- Anasphere SLWC #7

Probability Density / SLWC (g/m$^3$)

**10/13/17 PDF of Anasphere SLWC Values**

- Anasphere SLWC #9
- Anasphere SLWC #10

Probability Density / SLWC (g/m$^3$)

**Fig. 2.** PDFs of four side-by-side comparison flights of Anasphere SLWC sondes.

**Supplement:**

We thank all of the reviewers for their valuable comments. Below, we have added some responses (in red) to the comments submitted (in black).

**Reviewer 1 General Comments:**

The manuscript approaches the subject as if its TBS is a novel sampling technique that has never before been attempted, when in fact, the literature contains references to cloud sampling with tethered balloons for nearly the past 40 years. Kitchen and McClatchey (1981) flew an instrumented aerostat in cumulus clouds in the UK. Several investigators have followed suit, including measurements in the Arctic (Lawson et al. 2011) and at the South Pole (Lawson and Gettelman 2014). The manuscript needs to provide context for their research by discussing the previous efforts and explain why their current approach is an improvement, or how it differs.

We thank the reviewer for their comments, but believe historical TBS flight activities are discussed significantly in the existing manuscript on pg. 2, lines 16-26. We did add a reference to the highly relevant work discussed in Lawson 2011 and have included a summary of that publication to the existing discussion on pg. 2. We agree that we should better explain why our current approach is an improvement or how it differs. Towards that end we have revised the following text to read: "The work discussed herein pertains to the new capability of using tethered balloon systems within restricted airspace for persistent, repeatable, interannual flights inside Arctic mixed phase clouds, with supercooled liquid water sondes and distributed temperature sensing optical fiber systems."

The methodology used to process and present the measurements does not do justice to the larger variance in the different techniques being compared. While the variance can be seen in some of the measurements (e.g. time series in Fig. 3; scatterplots in Fig. 7), the overall message presented is that there is relatively good agreement in the measurements and that therefore they are useful. In fact, it is not at all clear that the vibrating wire technique is of any value in quantitatively measuring SLWC. The manuscript compares mean values of SLWC measured from the two vibrating wire techniques to validate the instruments when the variance in the measurements is larger than the mean; this is not an acceptable method for validating instrument performance. I agree that the technique may be useful as an SLWC detection method, and perhaps it can differentiate between low values i.e., < a few tenths g m-3 and high values (> ~ 1 g m-3), but even this cannot be validated since high SLWC measurements are not presented.

Again we thank the reviewer, but believe the discrepancies in the vibrating wire measurements are thoroughly discussed in the manuscript. In order to allow the reader to judge the efficacy of the vibrating wire SLWC measurements, comparisons between the SLWC measured by more than one vibrating wire instrument are shown, in addition to how the vibrating wire-measured SLWC compares with radiosonde-derived SLWC and MWR-derived SLWC. We admit we are unclear what the viewer is referring to by "The manuscript compares mean values of SLWC measured from the two vibrating wire techniques to validate the instruments . . . " since no comparison of mean values of SLWC measured from the two vibrating wire techniques as a validation of the instruments is presented. The mean differences between the pairs of Anasphere sondes in Figure 3 is presented to depict the noise or relative uncertainty in the Anasphere sonde measurements. If the reviewer is referring to Figure 9, which depicts LWP measured by the two different vibrating wire sondes and the MWR, the closest relationship between the mean and variance of the SLWC values occurs for the Anasphere sonde during the first descent with values of 0.02 and 0.0007 g/m$^3$ for the mean and variance, respectively.

I do feel that the DTS measurements are sufficiently accurate to be useful, but the validation is against a radiosonde. If the comparison is good, why not just use radiosonde data? Certainly it is more cost effective than the complexity involved with launching a tethered aerostat or helikite. One possible use of the DTS

measurements is to better define low-level inversions, but I did not find a meteorological use for this presented in the paper.

This is a good suggestion and the authors have added the following text to the Introduction in regard to the advantages of using tethered balloon DTS measurements compared to radiosonde data. "DTS provides greatly improved spatial and temporal resolution of temperature compared to radiosonde measurements and allows measurements to be collected continuously between the balloon and the surface for the duration of the tethered balloon flight."

Before the manuscript is fit for publication it should be modified considerably. The measurements are presented in a manner that embellishes their actual efficacy. The manuscript should present the data in an biased manner and focus on the ability of the TBS to make routine, long-term measurements, but eliminate the impression that the instruments themselves are useful for making quantitative measurements.

We thank the reviewer for their comments but believe the ability of the TBS to make routine, long-term measurements is discussed significantly in the Introduction and sections 2.1.2, 3.1.4, 3.1.5, 3.2.2, and 3.2.4 of the manuscript. We also believe the comparisons of the SLWC vibrating wire sonde data between both sondes, the MWR, and simultaneous radiosonde data are fairly presented and the discrepancies associated with the vibrating wire sonde measurements are extensively discussed in sections 3.1.1 – 3.1.3.

**Reviewer 1 Specific Comments:**

Introduction: Add TBS background references and discuss.

Please see response to Reviewer 1 General Comments.

Page 3, Lines 11 – 14: This is incorrect. There are much larger helikytes currently in service. For example, Bodenschatz, in Goettingen has deployed a 250 m3 helikyte from a ship in the Arctic.

We are uncertain what the reviewer is referring to in this case, since pg 3, lines 11-14 of the manuscript discuss the MWR. Perhaps the reviewer is referring to pg 3, line 31 – 32: "Helikites are typically used for flights with desired altitudes to 700 m Above Ground Level (AGL),a maximum payload of less than 10 kg, and in surface wind speeds less than 11 m/s." For clarity the authors have revised this line to begin with "In the ARM TBS, helikites . . . "

Figure 3: The figure and text states that the difference in mean values of SLWC between the two Anasphere devices is 0.01, 0.02 and 0.06 g m-3. This is meaningless without also giving some measurement of the variance, standard deviation or show a scatterplot. Even two noise signals can have the same mean. The figure suggests that there is little correlation between the two devices.

We thank the reviewer and have added pdfs of each sonde's measurements as suggested by Reviewer #2 to better illustrate the data collected by the two devices.

Figure 7: I applaud the authors for showing the scatterplots in this figure, but the interpretation of the results does not fit the data. I am not sure how an R2 value is even relevant. The scatterplot and time series tell the story. There is virtually no correlation in the measurements. The best that one can ascertain here is that the TBS is an SLWC detector.

As discussed in pg 12, line 18 through pg 13, line 9 the authors view the timing of SLWC detection between the airborne SLWC sensor and the surface-based MWR as significant. This result is especially relevant considering the uncertainties associated with the ceilometer-reported cloud base, the GPS-reported altitude of the airborne SLWC sonde in relation to the cloud base, and the determination of the cloud top altitude from the KAZR data, in addition to the variation in the cloud base on 10/15/16, and the saturation of the vibrating wire on the sonde and resulting ice shedding events on 10/20/16. As the reviewer mentions, the authors have included scatter plots of the results so the reader can evaluate the relationship between the two measurements.

Figures 8 and 9: This is a good representation of the measurements, but again, the interpretation is off the mark. The statement on p. 16, "Overall, the two sensors seem to provide broadly similar SLWC profiles, but not without some discrepancies. This increases confidence in the use of each of them and provides independent verification of the measurements." grossly overstates the efficacy of the measurements. The profiles in Figure 9 often diverge and differ by factors of 2 to 3. The authors need to show measurements with better correlation over a much larger range of SLWC's, and the sensors need to agree with SLWCad to convince readers that the instruments can be used quantitatively. Several aircraft campaigns have shown that single-layer ice-free Arctic stratus clouds typically have adiabatic SLWC profiles up to 60 to 90% of the distance between cloud base and cloud top. Of course, this will be challenging if the balloons cannot achieve higher altitudes.

The reviewer is correct in that there are differences in the magnitude of the SLWC detected by the various sensors, but this is to be expected for some of the reasons detailed on pg 16, lines 5-12. To this we have also added "The ice accumulation on the vibrating wire of each sonde may be dependent upon the upwind or downwind orientation of the sonde with respect to the balloon tether". The measurement of SLWC is a difficult one to make and these sensors are designed to be as inexpensive and lightweight as possible so as to be carried on disposable balloons, therefore the fact that the sensors roughly track each other and are generally within a factor of 2 is a reasonable achievement. If this was a comparison between aircraft instruments then we would expect better agreement, but one should not expect such accuracy from inexpensive disposable instruments.

Page 17, Lines 4 – 5: Equation 3 computes the adiabatic maximum SLWC. Since the vibrating wires measured a higher value than adiabatic, the measurements exceed the theoretical maximum. Also, once again, only reporting means does not show the variance in the measurements, which should also be reported, preferably by showing a scatterplot.

The theoretical calculation of adiabatic SLWC results in mean SLWC values of $< 0.05$ g/m$^3$ for 91% of 43 in-cloud TBS flights, which is inconsistent with in situ SLWC measurements within mixed-phase Arctic clouds from the literature. The in situ measurements collected by the TBS SLWC sondes exhibit better agreement with the literature values. This result depicts the value of the TBS to collect in situ atmospheric measurements and better refine cloud parameterization schemes used in climate models. A pdf and cdf of the SLWC data has also been added and the text updated to read "The probability density and cumulative distribution functions of all SLWC data collected with the Anasphere sondes on the TBS for the 43 flights are shown in Figure 12, and the probability density of SLWC values $< 0.05$ g/m$^3$ was 34%."

[Figure]

Page 17, Line 8: I have examined the Sand et al. (1984) article and I cannot find any text or figures that support the claim in the manuscript. Please show how 36% of the samples < 0.05 g m-3 was derived from the Sand paper. Also, the minimum measurement sensitivity of the J-W used in the Sand study was 0.1 g m-3 due to drift and noise affecting the baseline. The FSSP was used for lower values, but its fundamental measurement is drop size and raising drop size to the third power increases sizing errors commensurately. Thus, the FSSP is not a fundamental measurement of LWC. Reference to the Sand paper is a stretch and appears to be an unsubstantiated attempt to validate the vibrating wire measurements.

We thank the reviewer for this close review. On closer inspection the results referred to were in fact modified from Sand et al. (1984) and are reported at www-das.uwyon.edu/~geerts/cwx/notes/chap08/moist_cloud.html. We have updated the text and references to refer to the University of Wyoming reference.

Conclusions: The conclusions are very misleading. The measurements are limited to very low values of SLWC, so the dynamic range over which the measurements are constrained is tiny. Means are compared within this very limited dynamic range, making the comparison look good (e.g., the means of the measurements compared within 0.1 g m-3). This a totally unacceptable methodology for presenting the measurements. Showing the time series comparisons and scatterplots is representative and these figures are the saving grace of this manuscript. The paper should focus on the mechanics of how the TBS systems are operated, and that they can be successfully operated for extended periods of time. The vibrating wire technology is used on commercial aircraft to detect icing conditions, not to quantify icing rate, even though there was an attempt in the 80's to do so (i.e., Baumgardner and Rodi 1989). While Baumgardner and Rodi reported that independent laboratory calibration of the Rosemount icing probe improved its performance, they also reported that each of the four instruments tested had different mass sensitivities. There has been virtually no quantitative reporting of SLWC from an airborne Rosemount Icing probe in the literature. In time perhaps better SLWC measurement techniques applicable to TBS will be developed, implemented and reported.

As we have added the cdf and pdf of all SLWC values at the good suggestion of the reviewer, the reader can now observe that 40% of the SLWC data collected were >= 0.1 g/m$^3$. The text of the Conclusion has also been updated to refer to the SLWC data in the pdf/cdf figure.

**Reviewer 2 Comments:**

Page 1, Line 14: Please provide a bit more clarification of which kind of in situ atmospheric measurements you did.

We agree that the types of in situ measurements collected should be expounded. We have updated the text to read "of aerosol properties, cloud microphysical information, and thermodynamic structure".

Page 6, Figure 2: The beginning of the second sentence in the caption seems to be redundant. If I understood it correctly, the Anasphere SLWC sonde next to the InterMet radiosonde is shown on the left, the Anasphere SLWC sonde above the Anasphere tethersonde is shown in the middle and the Reading SLWC sonde is shown on the right. I would ask you to reformulate the caption accordingly.

We agree that this caption was poorly written and have revised it to read "Anasphere SLWC sonde left of InterMet radiosonde on TBS tether (left). Anasphere SLWC sonde above Anasphere tethersonde (center). InterMet radiosonde in center with Anasphere SLWC sonde on left and Reading SLWC sonde on right (right). "

Page 7, Figure 3: Figure 3 provides a good qualitative comparison of the sonde pairs, but I would recommend to include an additional figure to compare the measured distribution functions of SLWC. For that I would use only those times where both sondes had valid, non-zero readings, calculate the cumulative distribution function (or probability distribution function, whatever you like better) for each sonde and put the CDFs / PDFs for each pair of sondes in one plot.

We appreciate this feedback and have added the recommended PDF figure.

[Figure]

Page 9, Line 10: I am pretty sure that you mean Fig. 4 instead of Fig. 7 for the figure reference.

Corrected, thank you.

Page 11: I do not see a figure reference to Fig. 6.

Thank you, a reference has been added.

Page 12, Figure 6: There is almost no contrast in the data points showing the SLWC. I would suggest plotting the values below the detection limit either white or transparent to enhance the contrast for the data that actually matter.

We agree the contrast is insufficient and the figure has been replotted with SLWC values below the noise threshold of 0.02 g/m³ shown in grey.

Page 14: I do not see a figure reference to Fig. 8.

Thank you, the reference is in line 25 of page 14, "Figure 8 shows the time series of the flight and Figure 9 compares the two SLWC profiles."

Page 17: I do not see a figure reference to Fig. 10, Fig. 11 and Fig. 12.
Thank you, these three figure references have been added.

Page 19: I do not see a figure reference to Fig. 13.

Thank you, a reference has been added.

Page 20: I do not see a figure reference to Fig. 14.

Thank you, a reference has been added.

Page 20, Line 14: Please describe the radiation correction of the DTS in more detail (reference or equation, if possible). It might be best to explain the radiation correction in Subsection 2.3.

We appreciate the need for further detail and have expanded the existing section to include the text "While the multimode fiber used for TBS DTS measurements is white, some excess heating due to solar radiation could still occur. The iMet radiosonde temperatures are corrected within the collection software Skysonde, which was developed by the National Oceanic and Atmospheric Administration (NOAA), for solar radiation based on solar elevation and flight altitude. The correction factors were derived from a proprietary report developed by InterMet for NOAA (e.g., InterMet, 2009), and are not fixed but are interpolated between the solar elevations and altitudes shown in Table 2. In an effort to correct DTS temperatures for excess solar heating, the linear fit between the radiation-corrected iMet radiosonde temperatures and DTS temperatures was applied to the DTS temperature values for each flight. The mean RMSE between the iMet radiosonde temperatures and uncorrected DTS temperatures was 0.39 °C, improving to 0.32 °C after the corrections were applied to DTS temperatures." A table was also added of the correction factors that are applied to the iMet radiosonde temperatures.

Page 21, Line 10: When looking closely at Fig. 15, I do not see an increase of the temperature with altitude by more than 1 to 1.5 degrees Celsius. Please clarify where exactly the 3-4 _C warmer layer is located. Could it be the wrong figure that is put in the manuscript?

Thank you for this close review. An error was made with the color mapping and the map has been revised to better contrast the temperatures. The text has also been updated to read "Figure 16 depicts results from 7/10/18, when the continuous DTS temperature profiles and iMet radiosonde temperatures reveal a cooler layer at the surface below 100 m with a 1-1.5 °C warmer layer between 150 and 800 m, then another cooler layer above the inversion from 800 m to 1 km. The AMF3 radiosonde launch at 23:30 measured a similar temperature profile. The particle concentration measured by the POPS at a sample rate of 1 Hz demonstrates increased particle concentration within the temperature inversion, with fewer particles above the inversion and in the surface-cooled layer. The surface layer warmed in the afternoon and the base of the inversion layer became higher in altitude with time. An inversion was no longer present at 01:00 GMT, and the boundary layer became warmer and more well-mixed below 1 km. The particle concentrations measured by the POPS after 01:00 GMT were also similar at both measurement altitudes, which indicated the well-mixed boundary layer in the afternoon."

Page 21: I do not see a figure reference to Fig. 15 and Fig. 16.

Thank you, references have been added.

Page 22, Fig. 15 and 16: It is hard to actually see where the inversion is located in the vertical temperature profile. Would it be more intuitive to localize if you plot the potential temperature instead of the temperature? And is there an explanation for the high near-surface temperatures in Fig. 16 around 19:40 UTC?

Thank you again for this close review. The color mapping on this figure has been improved and the text has been revised to read "On 7/11/18 the surface layer below 200 m was roughly 2 °C cooler than on the previous day, as were temperatures in the inversion layer between 200 m and 1.2 km (Figure 17). POPS particle concentrations were elevated within the inversion layer and were similar to the observation on the previous day. Unlike the previous day, particle concentrations did not decrease to almost 0 above the inversion layer, indicating a less stratified aerosol profile. The base of the inversion layer decreased between 18:30 and 19:30, and a shallow ~50 m-deep warm layer was isolated around 400 m after 19:30. An iMet radiosonde on the tether corroborated this shallow warm layer measured by the DTS temperature profiles. No clouds were present within the TBS flight altitudes on either day. Elevated temperatures at the surface were caused by friction of the fiber against sharply-angled metal tubing as it entered and exited the calibration bath."

Page 23: I do not see a figure reference to Fig. 17.

Thank you, a reference has been added.

---

## Author Comment (AC2) · 11 Jul 2019

We thank all of the reviewers for their valuable comments. Below, we have added some responses to the comments submitted.

1) Page 1, Line 14: Please provide a bit more clarification of which kind of in situ atmospheric measurements you did.

Response 1) We agree that the types of in situ measurements collected should be expounded. We have updated the text to read "of aerosol properties, cloud microphysical

information, and thermodynamic structure".

2) Page 6, Figure 2: The beginning of the second sentence in the caption seems to be redundant. If I understood it correctly, the Anasphere SLWC sonde next to the InterMet radiosonde is shown on the left, the Anasphere SLWC sonde above the Anasphere tethersonde is shown in the middle and the Reading SLWC sonde is shown on the right. I would ask you to reformulate the caption accordingly.

Response 2) We agree that this caption was poorly written and have revised it to read "Anasphere SLWC sonde left of InterMet radiosonde on TBS tether (left). Anasphere SLWC sonde above Anasphere tethersonde (center). InterMet radiosonde in center with Anasphere SLWC sonde on left and Reading SLWC sonde on right (right). "

3) Page 7, Figure 3: Figure 3 provides a good qualitative comparison of the sonde pairs, but I would recommend to include an additional figure to compare the measured distribution functions of SLWC. For that I would use only those times where both sondes had valid, non-zero readings, calculate the cumulative distribution function (or probability distribution function, whatever you like better) for each sonde and put the CDFs / PDFs for each pair of sondes in one plot.

Response 3) We appreciate this feedback and have added the recommended PDF figure.

4) Page 9, Line 10: I am pretty sure that you mean Fig. 4 instead of Fig. 7 for the figure reference.

Response 4) Corrected, thank you.

5) Page 11: I do not see a figure reference to Fig. 6.

Response 5) Thank you, a reference has been added.

6) Page 12, Figure 6: There is almost no contrast in the data points showing the SLWC. I would suggest plotting the values below the detection limit either white or transparent

to enhance the contrast for the data that actually matter.

Response 6) We agree the contrast is insufficient and the figure has been replotted with SLWC values below the noise threshold of 0.02 g/m3 shown in grey.

7) Page 14: I do not see a figure reference to Fig. 8.

Response 7) Thank you, the reference is in line 25 of page 14, "Figure 8 shows the time series of the flight and Figure 9 compares the two SLWC profiles."

8) Page 17: I do not see a figure reference to Fig. 10, Fig. 11 and Fig. 12.

Response 8) Thank you, these three figure references have been added.

9) Page 19: I do not see a figure reference to Fig. 13.

Response 9) Thank you, a reference has been added.

10) Page 20: I do not see a figure reference to Fig. 14.

Response 10) Thank you, a reference has been added.

11) Page 20, Line 14: Please describe the radiation correction of the DTS in more detail (reference or equation, if possible). It might be best to explain the radiation correction in Subsection 2.3.

Response 11) We appreciate the need for further detail and have expanded the existing section to include the text "While the multimode fiber used for TBS DTS measurements is white, some excess heating due to solar radiation could still occur. The iMet radiosonde temperatures are corrected within the collection software Skysonde, which was developed by the National Oceanic and Atmospheric Administration (NOAA), for solar radiation based on solar elevation and flight altitude. The correction factors were derived from a proprietary report developed by InterMet for NOAA (e.g., Inter-Met, 2009), and are not fixed but are interpolated between the solar elevations and altitudes shown in Table 2. In an effort to correct DTS temperatures for excess solar

heating, the linear fit between the radiation-corrected iMet radiosonde temperatures and DTS temperatures was applied to the DTS temperature values for each flight. The mean RMSE between the iMet radiosonde temperatures and uncorrected DTS temperatures was 0.39 °C, improving to 0.32 °C after the corrections were applied to DTS temperatures." A table was also added of the correction factors that are applied to the iMet radiosonde temperatures.

12) Page 21, Line 10: When looking closely at Fig. 15, I do not see an increase of the temperature with altitude by more than 1 to 1.5 degrees Celsius. Please clarify where exactly the 3-4 _C warmer layer is located. Could it be the wrong figure that is put in the manuscript?

Response 12) Thank you for this close review. An error was made with the color mapping and the map has been revised to better contrast the temperatures. The text has also been updated to read "Figure 16 depicts results from 7/10/18, when the continuous DTS temperature profiles and iMet radiosonde temperatures reveal a cooler layer at the surface below 100 m with a 1-1.5 °C warmer layer between 150 and 800 m, then another cooler layer above the inversion from 800 m to 1 km. The AMF3 radiosonde launch at 23:30 measured a similar temperature profile. The particle concentration measured by the POPS at a sample rate of 1 Hz demonstrates increased particle concentration within the temperature inversion, with fewer particles above the inversion and in the surface-cooled layer. The surface layer warmed in the afternoon and the base of the inversion layer became higher in altitude with time. An inversion was no longer present at 01:00 GMT, and the boundary layer became warmer and more well-mixed below 1 km. The particle concentrations measured by the POPS after 01:00 GMT were also similar at both measurement altitudes, which indicated the well-mixed boundary layer in the afternoon."

13) Page 21: I do not see a figure reference to Fig. 15 and Fig. 16.

Response 13) Thank you, references have been added.

14) Page 22, Fig. 15 and 16: It is hard to actually see where the inversion is located in the vertical temperature profile. Would it be more intuitive to localize if you plot the potential temperature instead of the temperature? And is there an explanation for the high near-surface temperatures in Fig. 16 around 19:40 UTC?

Response 14) Thank you again for this close review. The color mapping on this figure has been improved and the text has been revised to read "On 7/11/18 the surface layer below 200 m was roughly 2 °C cooler than on the previous day, as were temperatures in the inversion layer between 200 m and 1.2 km (Figure 17). POPS particle concentrations were elevated within the inversion layer and were similar to the observation on the previous day. Unlike the previous day, particle concentrations did not decrease to almost 0 above the inversion layer, indicating a less stratified aerosol profile. The base of the inversion layer decreased between 18:30 and 19:30, and a shallow ∼50 m-deep warm layer was isolated around 400 m after 19:30. An iMet radiosonde on the tether corroborated this shallow warm layer measured by the DTS temperature profiles. No clouds were present within the TBS flight altitudes on either day. Elevated temperatures at the surface were caused by friction of the fiber against sharply-angled metal tubing as it entered and exited the calibration bath."

15) Page 23: I do not see a figure reference to Fig. 17.

Response 15) Thank you, a reference has been added.

Please also note the supplement to this comment:
https://www.atmos-meas-tech-discuss.net/amt-2019-117/amt-2019-117-AC2-supplement.pdf

[Figure]

**Fig. 1.** PDFs of four side-by-side comparison flights of Anasphere SLWC sondes.

**7/10/18 - 7/11/18 TBS DTS at AMF3 with iMet Radiosonde Temperatures and POPS Particle Concentration**

**Fig. 2.** 7/10/18 – 7/11/18 TBS DTS profiles at AMF3 with TBS iMet temperatures (squares), free-flight radiosonde temperatures (diamonds), and POPS particle concentrations (circles).

**7/11/18 TBS DTS at AMF3 with iMet Radiosonde Temperatures and POPS Particle Concentration**

**Fig. 3.** 7/11/18 TBS DTS profiles at AMF3 with TBS iMet temperatures (squares) and POPS particle concentrations (circles).

**SLWC Values from TBS Anasphere Sondes from 2016 - 2017**

**Fig. 4.** TBS Flight of two SLWC sondes with concurrent free balloon radiosonde launch at 10/13/17 23:27 UTC.